# Structural basis for the E3 ligase activity enhancement of yeast Nse2 by SUMO-interacting motifs

Nathalia Varejão [1,3], Jara Lascorz [1,3], Joan Codina-Fabra [2], Gemma Bellí [2], Helena Borràs-Gas[1], Jordi Torres-Rosell [2] & David Reverter [1✉]

Post-translational modification of proteins by ubiquitin and ubiquitin-like modifiers, such as SUMO, are key events in protein homeostasis or DNA damage response. Smc5/6 is a nuclear multi-subunit complex that participates in the recombinational DNA repair processes and is required in the maintenance of chromosome integrity. Nse2 is a subunit of the Smc5/6 complex that possesses SUMO E3 ligase activity by the presence of a SP-RING domain that activates the E2~SUMO thioester for discharge on the substrate. Here we present the crystal structure of the SUMO E3 ligase Nse2 in complex with an E2-SUMO thioester mimetic. In addition to the interface between the SP-RING domain and the E2, the complex reveals how two SIM (SUMO-Interacting Motif) -like motifs in Nse2 are restructured upon binding the donor and E2-backside SUMO during the E3-dependent discharge reaction. Both SIM interfaces are essential in the activity of Nse2 and are required to cope with DNA damage.

[1] Institut de Biotecnologia i de Biomedicina (IBB) and Dept. de Bioquímica i Biologia Molecular, Universitat Autònoma de Barcelona, 08193 Bellaterra, Spain. [2] IRBLLEIDA, Dept. Ciències Mèdiques Bàsiques, Universitat de Lleida, Lleida, Spain. [3] These authors contributed equally: Nathalia Varejão, Jara Lascorz. ✉email: david.reverter@uab.cat

Post-translational modification of proteins by ubiquitin (Ub) and ubiquitin-like proteins (UbL) is achieved via an orchestrated pathway conducted by the E1 activating enzyme, the E2 conjugating enzyme, and E3 ligase enzyme[1,2]. SUMO is a Ubiquitin-like modifier that participates in a wide plethora of functions inside the cell, including DNA damage repair pathways[3,4]. Humans possess three conjugatable SUMOs: SUMO2 and SUMO3 share 97% sequence identity, and both share about 50% identity with SUMO1. The *Saccharomyces cerevisiae* genome only contains one SUMO molecule, named Smt3. Once the E1 is charged with SUMO in an ATP-dependent manner forming a thioester bond with an internal cysteine residue, SUMO is sequentially transferred to a cysteine residue of the E2 forming the E2~SUMO thioester, which ultimately can be discharged on a lysine residue of the substrate forming an isopeptide bond[5]. However, the SUMO transfer from E2 to the substrate can be enhanced by the action of E3 ligases, which facilitate substrate binding while increasing the catalytic rate for the E2~SUMO discharge on the substrate. Such E3-dependent stimulation of the E2~SUMO thioester discharge is mechanistically conducted by the stabilization of a closed or active conformation of the E2~SUMO when binding the E3 ligase[6]. Structures of ubiquitin and Nedd8 E3 ligases in complex with charged E2 thioester also support a similar mechanism of stabilization of the closed conformation[7,8], thus indicating a general mechanism for all E3 ligases albeit the specific contacts established in each particular case. In some instances, E2s can also employ a similar mechanism to stabilize a closed conformation in the absence of E3[9].

The ubiquitin conjugation pathway contains a large family of encoded E3 ligases in the genome[10], highlighting its major role in the pathway by conferring substrate specificity and enhancing the isopeptide bond formation. However, in SUMO only a few bona fide E3 ligases have been discovered so far. The yeast Siz (PIAS in humans) and Nse2 (or Mms21) family members contain an SP-RING domain, similar to the ubiquitin RING-E3 ligases, that binds charged E2s[6]. RanBP2 constitutes a particular class of SUMO E3 ligases, not present in ubiquitin, containing an IR1-M-IR2 motif that binds the charged E2 via a SIM motif (SUMO-Interacting Motif) present in each internal repeat (IR)[11]. ZNF451 (human) is a recently well-characterized SUMO E3 ligase containing a catalytic module with two consecutive SIM motifs involved in the SUMO E3 ligase mechanism[12,13], one for binding the charged E2~SUMO and another interacting a second SUMO at the backside of the E2[14]. SIMs are short sequence motifs containing a stretch of three to four hydrophobic residues bordered by acidic residues that bind in a β conformation the SUMO β-sheet in a parallel or antiparallel orientation[15,16]. The structure of the RING-type E3 ligase Siz1 in complex with the charged E2~SUMO and PCNA also revealed the presence of a SIM-like interaction motif binding the charged SUMO to enhance SUMO discharge[17], thus confirming the participation of SIM motifs in all known SUMO E3 ligases[18]. SIM motifs constitute the major class of non-covalent SUMO binders and have also been reported to be present in many proteins for recruitment of SUMO modules involved in a plethora of functions[19–21].

Structural and functional results have recently revealed a relevant role of a second SUMO binding the charged E2 backside during the E3-dependent conjugation reaction[14,17,22–24]. Non-covalent SUMO binding to the Ubc9 backside is equivalent to the ubiquitin-E2-backside interaction described for different E2s[25–27] and in both instances, SUMO and ubiquitin, they have been reported to increase the rate of SUMO or ubiquitin chain formation[12,22–25,28,29]. The SUMO-Ubc9 backside interaction has been reported to be essential in the E3 activity of ZNF451, and to increase the catalytic efficiency in Siz1[14,17]. Also, non-covalent

binding to the E2 backside plays an important role in the RING E3-ligase catalyzed ubiquitin discharge reaction, at least for UbcH5[30]. In this instance, an allosteric mechanism has been proposed for ubiquitin-binding to the UbcH5 backside to stabilize the charged E2-E3 in a closed and optimal conformation to stimulate ubiquitin transfer[30,31].

We are pursuing to decipher the molecular mechanisms behind the SUMO E3 ligase activity of Nse2, which is one of the few bona fide SUMO E3 ligase discovered so far. Nse2 is a protein subunit embedded in the Smc5/6 complex, a large DNA-binding multi-subunit protein complex that belongs to the family of cohesin (Smc1/3) and condensin (Smc2/4) which are all involved in the maintenance of chromatin and chromosomal architecture and dynamics[32–35]. The structure of an Smc protein (Structural Maintenance of Chromosomes) consists of a head domain with ATPase activity, a long coiled-coil arm region, and a hinge domain that dimerizes with another Smc molecule. Additionally, six Nse subunits (Non-smc elements) interact with the Smc5 and Smc6 heterodimer, binding preferentially to the ATPase head domains, and creating the characteristic shapes of the Smc5/6 complex[36,37]. Different low-resolution cryoEM structures have been recently reported for the Smc5/6 complex[38,39]. Nse2 is a unique component of the Smc5/6 complex that binds the middle region of the coiled-coil Arm of Smc5 and possesses SUMO E3 ligase activity, moreover, the SUMOylation activity has been involved in DNA damage repair processes[40]. Although a minimal fragment of Smc5 coiled-coil bound to Nse2 is sufficient to provide SUMO E3 ligase activity in vitro[41], in vivo SUMO conjugation activity seems to be regulated in the context of the whole Smc5/6 complex, probably by large conformational changes induced by the ATPase activity of the head domains[42]. Also, we have recently reported the enhancement of the E3 ligase activity of Nse2 by direct binding to DNA, perhaps by an allosteric mechanism involving a rearrangement of the E3 ligase module[41].

Here, we present the crystal structure of the Smc5/Nse2 E3 ligase in complex with an E2-SUMO$_D$ thioester mimetic at 3.3 Å resolution (SUMO$_D$ refers to donor SUMO), which sheds light on the SUMO E3 ligase activity of Nse2 by revealing the atomic details of this multiple interface enzyme-substrate complex. In particular, we identify and investigate the role of the C-terminal SIM2 of yeast Nse2, which enhances the E3 ligase activity of Nse2 by anchoring a second SUMO$_B$ at the backside of Ubc9 (SUMO$_B$ refers to backside SUMO). The complex structure also reveals the combined action of two SIM-like motifs in the E3 activity of Nse2, SIM1 tethers the donor SUMO$_D$ and SIM2 contributes to anchoring SUMO$_B$ at the E2 backside. All these E3 interfaces contribute to the stabilization of an optimal or closed conformation of the charged E2~SUMO$_D$ thioester during the E3 ligase activity of Nse2, thus contributing to DNA damage repair in budding yeast.

## Results

**Structure of Nse2/Smc5 in complex with the E2-SUMO thioester mimetics.** To gain insights into the mechanism of SUMO E3 ligases, we have reconstituted the complex between yeast Smc5/Nse2 with an E2-SUMO thioester mimetic, which contains a stable peptide bond between E2 and SUMO, instead of the labile natural thioester bond. To facilitate comprehension, we will use SUMO to refer to yeast Smt3 and E2 to yeast Ubc9 throughout the text. Such E2-SUMO thioester mimetic has been engineered based on previous work[17,43], by the substitution of Ala129 to lysine in a location next to the active site Cys93 in Ubc9, and Lys153 for arginine to prevent unwanted E2 SUMOylation[29,44]. Under physiological pH conditions, Lys129 can nucleophilically attack the Cys93~SUMO thioester forming a

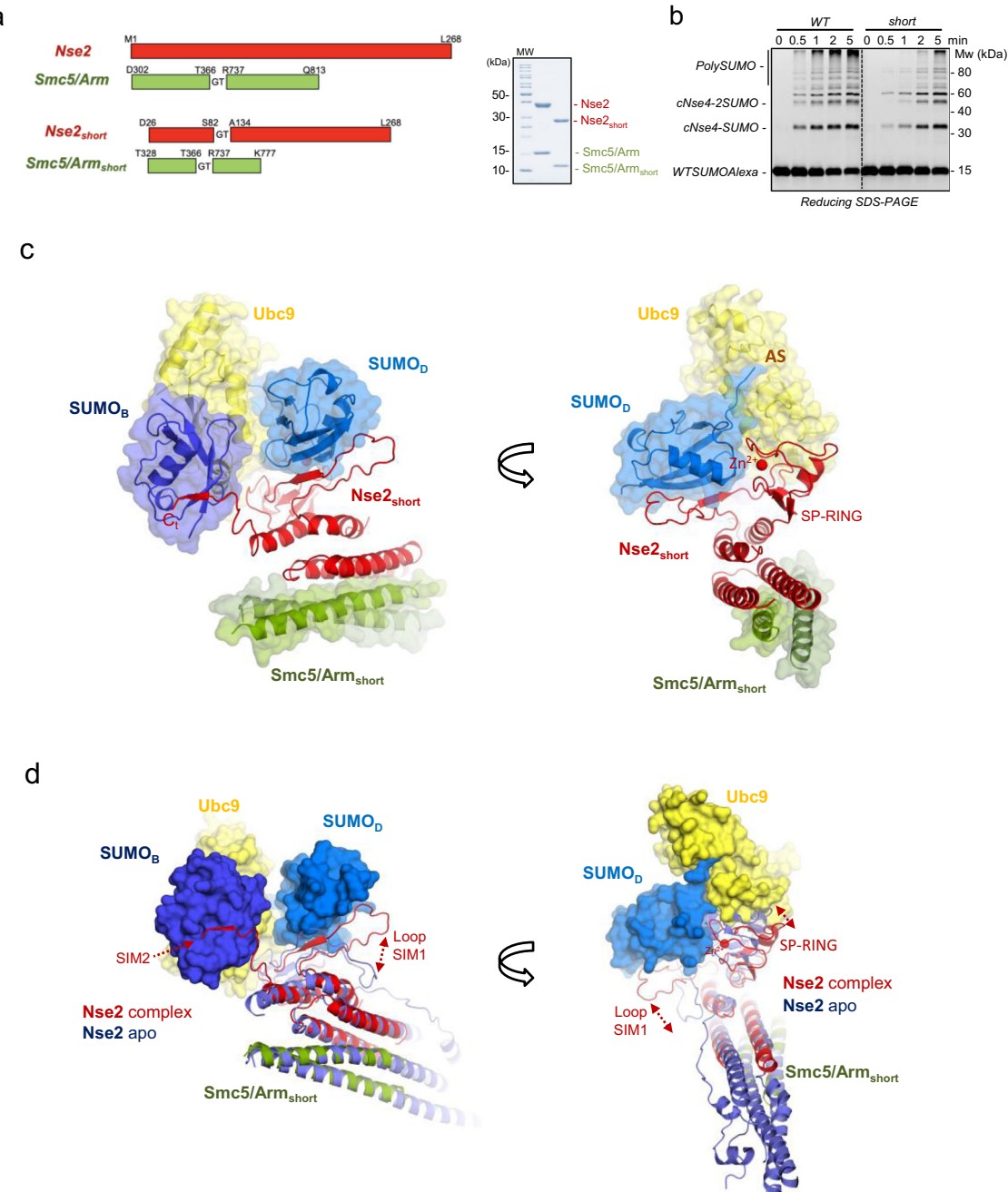

**Fig. 1 Structure of yeast Nse2/Smc5 in complex with the E2-SUMO thioester mimetics. a** Schematic representation of the engineered short form of Nse2 and Arm/Smc5 coiled-coil. **b** Multiple-turnover SUMOylation reactions using Nse2/Arm-Smc5 and $_{short}$Nse2/Arm-Smc5. SDS-PAGE represents one of three technical replicates. **c** Side views of the $_{short}$Nse2/Arm-Smc5 E2-SUMO$_D$ SUMO$_B$ complex. Each subunit is color coded. The crystal asymmetric unit contained one complex composed of Smc5/Nse2$_{short}$, Ubc9-SUMO$_D$ thioester mimetic and SUMO$_B$. **d** Side views of the structural alignment of $_{short}$Nse2/Arm-Smc5 E2-SUMO$_D$ SUMO$_B$ complex with Nse2/Arm-Smc5 apo complex (3HTK). Both structures were aligned using the Arm/Smc5 coiled-coils domains. The superimposition reveals a major conformational change of loop SIM1 to interact with SUMO$_D$ and the C-terminal tail of Nse2 (named SIM2) to interact with SUMO$_B$.

stable isopeptidic bond located in a similar position as the natural E2~SUMO$_D$ thioester bond.

Previous studies and our experimental data indicated that a SUMO$_B$ fused to the C-terminus of Nse2 can stabilize the complex through non-covalent interactions with the E2 backside surface. Our initial crystallization trials of Ubc9-SUMO thioester mimetic in complex with Smc5/Nse2 or fused Smc5/Nse2-SUMO$_B$ were unsuccessful. However, we engineered a shorter Nse2, containing the SP-RING domain bound to a shorter Smc5/Arm coiled-coil (Fig. 1a). The E3 ligase activity of the Smc5/Nse2$_{short}$ is comparable

to the WT Smc5/Nse2, at least for the single cNse4 conjugation (cNse4-SUMO), only high molecular conjugates are substantially diminished (Fig. 1b). Initially, crystals of Smc5/Nse2$_{short}$-SUMO$_B$ fusion in complex to E2-SUMO$_D$ thioester mimetics diffracted only to 5.5 Å. However, under similar conditions, the crystals diffracted up to 3.3 Å when non-fused Smc5/Nse2$_{short}$ was mixed with free SUMO$_B$ (Supplementary Table 1).

The asymmetric unit contained one complex composed by Smc5/Nse2$_{short}$, E2-SUMO$_D$ thioester mimetic, and SUMO$_B$ attached to the E2 backside (Fig. 1c). The structure unveils an

extended interaction between the charged E2-SUMO_D and the SP-RING domain of Nse2, including direct contacts with both E2 and SUMO_D. In addition to the SP-RING interaction, SUMO_D also forms an extended interface with a large loop of Nse2 (named Loop SIM1), which suffers a major conformational change forming a β-like interaction with SUMO_D, reminiscent of a SIM-like conformation (Fig. 1c, d). Strikingly, the C-terminal tail of Nse2 can be observed in the electron density maps forming a second SIM-like interaction (named SIM2) with SUMO_B, which is tethered to the Ubc9 backside. Thus, Nse2 clamps the SUMO_B-E2-SUMO_D substrate complex utilizing two SIM-like motifs, each one binding either the donor SUMO_D or the backside SUMO_B, respectively.

Overall, the structure of Smc5/Nse2 in complex with E2-SUMO_D thioester mimetic shares some features with the Siz1 E3 ligase complex, mainly in the SP-RING interface with Ubc9. However, major differences, unique to Nse2, arise in the interface between SIM1 and SIM2 motifs with SUMO_B-E2-SUMO_D. Alike the other SUMO E3 ligases, Siz1, Nup358, or ZNF451, SIM-like elements contribute to the stabilization of the closed conformation of the E2~SUMO thioester to enhance the SUMO discharge on the substrate.

**Role of a C-terminal SIM motif (SIM2) in the catalytic activity of Nse2.** The C-terminal tail of Nse2 was not observed in the yeast Smc5/Nse2 structure (PDB 3HTK)[45], despite the presence of a hydrophobic stretch resembling a SIM motif (Fig. 2a). Such SIM-like motif is not present neither in humans nor in fission yeast, indicating that it would pertain to some particular function in the clade of *S. cerevisiae* (Fig. 2a). Yeast and human SP-RING domains (PDBs 3HTK and 2YU4) can be superimposed (Cα rmsd 1.70 Å, identity 25.5%), but display different lengths of the C-terminal α-helix (Supplementary Fig. 1a). Our structure of Smc5/Nse2 in complex with the E2-SUMO thioester mimetic shows the electron density of the C-terminal tail SIM motif (SIM2) unequivocally bound to the backside SUMO_B (Figs. 1, 2b).

SIM2 interacts with SUMO_B in a canonical SIM-like conformation, burying the hydrophobic side chains of Ile264 and Val266 (positions 1 and 3 in SIM2) in a SUMO surface cavity formed between the α-helix and the β-sheet (Fig. 2c). Hydrophobic contacts are engaged with Phe37, Leu48, and Ala51 in the SUMO cavity, as well as electrostatic contacts with SUMO positive charges, namely between Asp265 (position 2 in SIM2) and His23 and, interestingly, between the C-terminal carboxylate group of Nse2 and Arg47 (Fig. 2c). The structure reveals the presence of two main chain hydrogen bonds between SIM2 and the SUMO β-strand, forming a characteristic parallel SIM β conformation. Interestingly, the structure of the Nse2 SIM2 overlaps very well with the structure of other SIM motifs, such as the C-terminal SIM2 of PIAS1, from the human SUMO1-SIM2 peptide structure (6V7P)[46] (Fig. 2d). It is particularly relevant to observe similar locations for Ile264 and Val266 in the SUMO cavity, despite the different orientations of the main chain (Fig. 2d).

To check the role of the C-terminal tail of Nse2 (SIM2) in the SUMO E3 ligase activity, three-point mutants of SIM2 were produced: I264A/V266A (second and fourth positions of SIM2); I264P (first position of SIM2); and V266R (third position of SIM2). Also, C-terminal deletion mutants included: Δ4C-, Δ8C- and Δ16C-Nse2, the latter removes the last two turns of the α-helix. To validate the stability of the deletion mutants, the crystal structure of the Smc5/Δ16C-Nse2 deletion was solved at 3.3 Å resolution (Supplementary Fig. 1b), displaying a similar structure as the wild type (Cαα rmsd between Δ16C and WT-Nse2 is 0.56 Å). Additionally, intrinsic Trp-fluorescence with the Smc5/Δ16C-Nse2

deletion displayed similar emission values and DNA-dependent red-shift as the WT-Nse2 (Supplementary Fig. 1c)[41]. This biophysical analysis confirmed that even in the largest C-terminal deletion mutant, Δ16C-Nse2, the structural integrity, and stability seem to be preserved.

In vitro assays using both human and yeast conjugation systems (SUMO, E1, and E2 enzymes, ATP) display a clear decrease in SUMO conjugation in all Nse2 mutants (Supplementary Fig. 2b). Since no stability problems were observed, we attribute this reduction in conjugation activity to a role of SIM2 during the E3 ligase activity of Nse2. Interestingly, under similar conditions, the yeast enzymes displayed a much higher conjugation activity than the human counterparts (Supplementary Fig. 2). All three SUMO substrates assayed, namely the Nse4 C-terminal Kleisin domain (_CNse4), the P53 C-terminal domain (_Cp53), and auto-conjugation on the Arm/Smc5, displayed a Nse2 E3-dependent activity (Fig. 2e–g, left), and in all cases, SUMO conjugation was diminished in all tested SIM2 mutants. The strongest decrease, around 80%, was observed for Δ4C-Nse2 (ΔSIM2), and around 30–50% for I264P and V266R (Fig. 2e–g). Altogether, these results reveal a role for SIM2 in the SUMO E3 ligase activity of Nse2.

**The SIM2 participates in DNA damage repair in yeast.** The Nse2 SUMO ligase cooperates with the Smc5/6 complex in genome integrity[40]. Thus, inactivation of the SIM2 is expected to decrease the efficiency of DNA repair in cells. To test the role of the SIM2 in vivo, we introduced a STOP codon just after Ala263 at the endogenous budding yeast NSE2 locus, thus preventing translation of the C-terminal IDVL SIM2 sequence (NSE2p.A263_I264X; hereafter referred to as nse2-SIM2Δ for simplicity). nse2-SIM2Δ cells did not have any apparent phenotype under normal conditions or after exposure to DNA damage, indicating that the SIM2 is not essential for normal growth or DNA repair in otherwise wild-type yeast cells (Fig. 3a). This may be due to the presence of compensatory mechanisms masking the role of the SIM2 through alternative DNA repair pathways. For example, SUMOylation of the Smc5 protein is not essential in yeast but cooperates with the Mms4-Mus81 and Slx4 structure-specific endonucleases, and the SUMO-like domain containing protein Esc2 for DNA alkylation damage repair[47]. Therefore, we combined the nse2-ΔSIM2 mutant with deletions in the MMS4, SLX4, and ESC2 genes. As shown in Fig. 3a, truncation of the Nse2 protein just before the SIM2 motif reduces the growth of slx4Δ and esc2Δ cells and aggravates the sensitivity to MMS of the mms4Δ, esc2Δ, and slx4Δ mutant cells. This suggests that the SIM2 in Nse2 acts synergistically with genes involved in DNA recombination intermediate processing to promote DNA repair in budding yeast.

Many smc5/6 mutants are also hypomorphic for Nse2-dependent sumoylation, most probably because of reduced Smc5/6 function[42]. We thus reasoned that truncation of SIM2 would synergize with mutations in other subunits of the complex, by further compromising Smc5/6 function. To explore this possibility, we crossed nse2-SIM2Δ cells with thermosensitive mutants affected in the NSE1, NSE3, NSE4, NSE5, or NSE6 subunits of the Smc5/6 complex. After sporulation, we selected double mutants and compared their growth to wild type and single mutant cells. As shown in Fig. 3b, truncation of the SIM2 in different smc5/6 mutant backgrounds reduced their growth at the permissive temperature and increased the thermosensitivity of cells. Overall, we conclude that the SIM2 in Nse2 cooperates with other Smc5/6 subunits and DNA repair pathways to promote repair of DNA damage and normal cell growth.

**The C-terminal SIM2 of Nse2 fixes SUMO_B to the E2 backside.** Our complex structure reveals the presence of a second SUMO_B bound to the backside of Ubc9. SUMO_B was added during

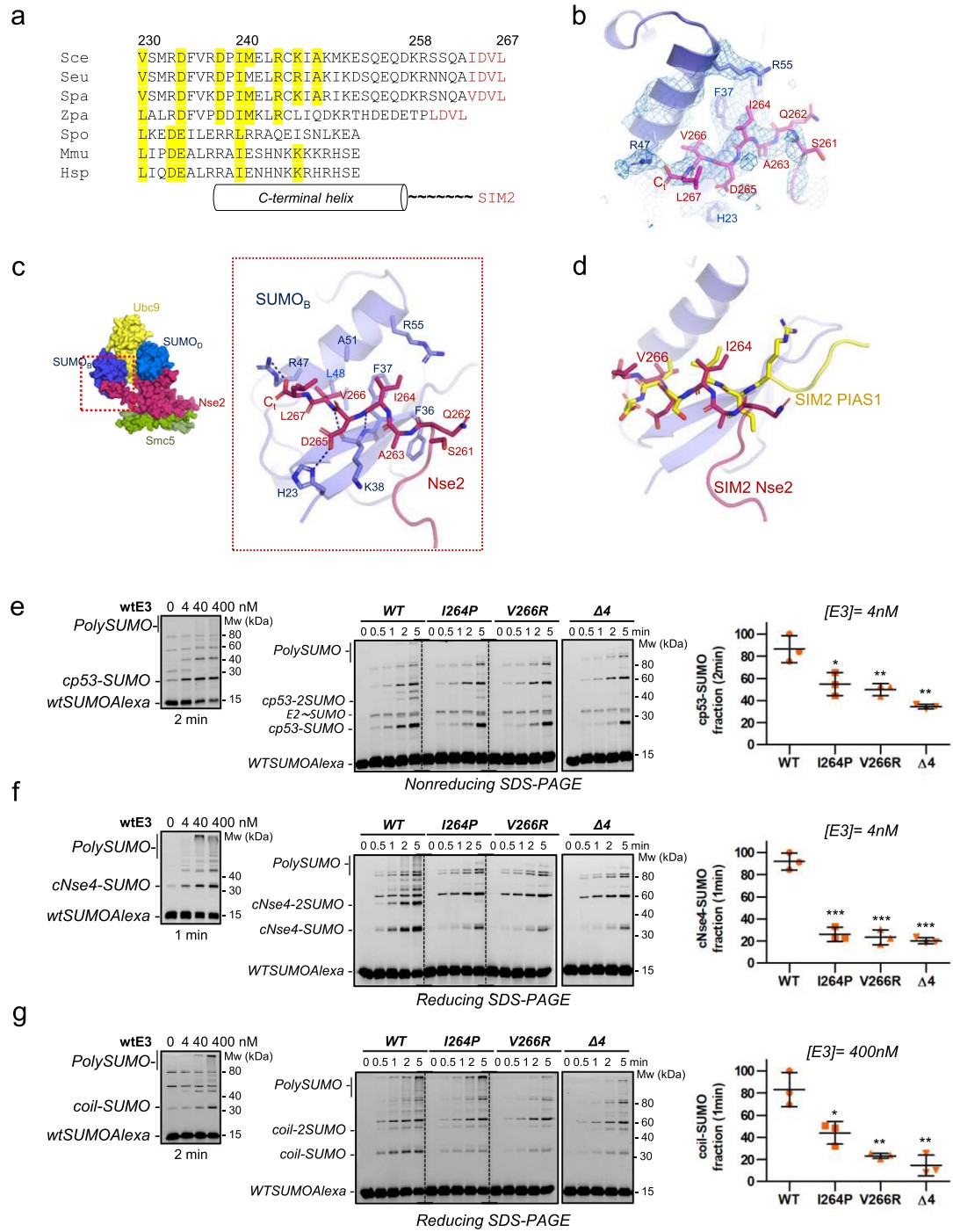

**Fig. 2 Role of a C-terminal SIM motif (SIM2) in the catalytic activity of Nse2. a** Multiple alignment of Nse2 sequences from *Saccharomyces cerevisiae*, *S. eubayanus*, *S. paradoxus*, *Zygos. parabailli*, *Schizos. pombe*, *Mus musculus* and *Homo sapiens* (in red SIM2). **b** Electron density maps of SIM2 Nse2 and SUMO_B. **c**. Stick representation of the complex between SIM2 and SUMO_B. **d** Structural alignment of Nse2 SIM2 and SIM2 from PIAS1-peptide complex (PDB 6V7P), evidencing similar locations of Ile264 and Val266 in the SUMO_B cavity. **e–g** Multiple-turnover SUMOylation reactions of yeast Nse2/Arm-Smc5 complex (wild type and mutants) using cP53, cNse4 and coil/Smc5 substrates. Enzyme concentration was chosen based on the concentration dependence assay (left). Error bars showing that point mutations or deletion of SIM2 reduce E3 activity (right). Data values represent the mean ±SD, $n = 3$ technical replicates. Significance was measured by a two-tailed unpaired t-test relative to wild type. All data were analyzed with a 95% confidence interval. *$P < 0.05$, **$P < 0.01$, ***$P < 0.001$. Exact $P$ values from the left to right: (**e**) 0.027, 0.009, 0.002; (**f**) 0.0003, 0.0003, 0.0001; (**g**) 0.021, 0.002, 0.002. Source data are provided as a Source Data file.

preparation of the complex, in contrast to the E3-SUMO_B fusion used in the Siz1 complex[17]. In our structure, SUMO_B displays two different interfaces, one side faces the backside of Ubc9, and the opposite side engages contacts with the C-terminal SIM2 of Nse2 (Fig. 1c). Non-covalent SUMO binding to the E2 backside has been reported by different groups, it was initially associated with the formation of SUMO chains, and now it seems to be required in the E3-dependent discharge reaction[12–14,17,22–24]. SUMO_B binds Ubc9 like other reported E2-SUMO structures, through an interface enriched with electrostatic interactions: Asp68 engaged

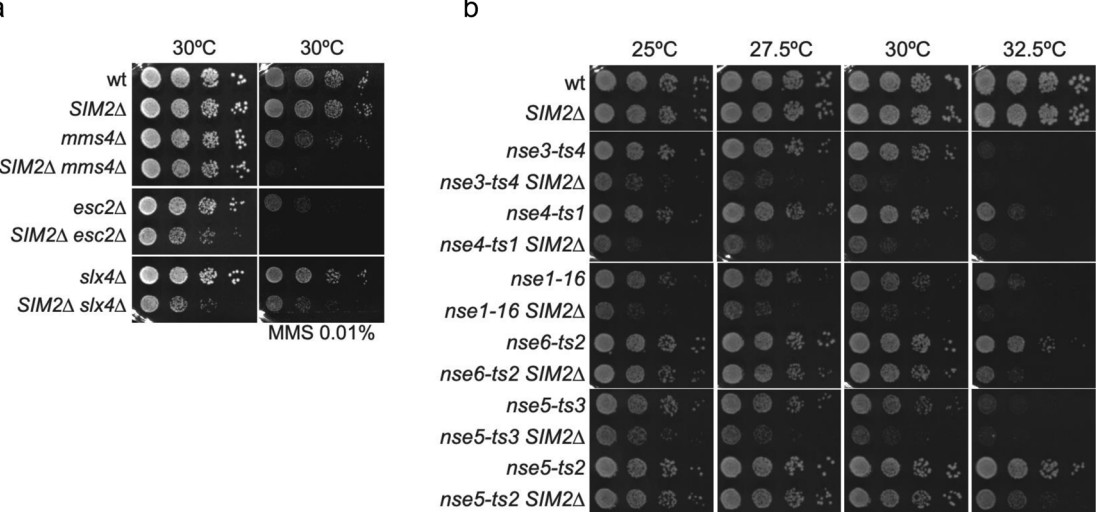

**Fig. 3 Analysis of *nse2-SIM2Δ* mutants in yeast. a** Growth test analysis of *nse2-SIM2Δ* (*SIM2Δ*) cells in combination with *mms4Δ*, *esc2Δ* and *slx4Δ*; 10-fold serial dilutions of liquid cultures were spotted in YPD in the presence or not of MMS 0.01% and incubated at 30 °C for 48h. **b** Growth test analysis of *nse2-SIM2Δ* (*SIM2Δ*) cells in combination with the indicated Smc5/6 complex alleles; 10-fold serial dilutions of liquid cultures were spotted in YPD and incubated at the indicated temperatures for 48h.

with Arg13 and Arg17 from the 1-helix of Ubc9; and Glu90 with His20 and the main chain oxygen of Gly23 from Ubc9 (Fig. 4a). Interestingly, E2 overlapping in the Nse2 and Siz1 complexes displays around 2 Å displacement of SUMO$_B$, resulting in a contact loss between Asp68 and arginine in the Siz1 complex (more than 5 Å apart) (Fig. 4b). Our hypothesis points to the role of SIM2 to anchor SUMO$_B$ to the E2 backside and to contribute to the formation of a full-contact interface.

A point mutant in the E2-interface in Smt3, D68R, can disrupt the E2-SUMO$_B$ interaction[22]. In vitro conjugation reactions with Smt3 D68R shows a strong decrease in the conjugation rates, particularly for polySUMO chains, either in the presence or absence of the E3 ligase (Fig. 4c). Despite the slow rate formation of E2~SUMO thioester with Smt3 D68R (Fig. 4d, left), as previously reported[22], single-turnover discharge reactions with extra WT or Smt3 D68R were conducted after stopping the E1 activation with EDTA. End-point reactions reveal an enhancement of the E2 discharge on $_C$Nse4 with extra WT Smt3, compared to Smt3-D68R, which shows similar conjugation rates as in the absence of E3 (Fig. 4d, upper). A similar trend was observed with $_C$p53, but in this instance, the E2 discharge is faster and completed after 30 s with WT Smt3, in contrast to the slower rates observed with Smt3-D68R (Fig. 4d, below). These results support an enhancement of the discharge reaction by the non-covalent E2-SUMO$_B$ interaction.

To establish a link between the C-terminal SIM2 and the SUMO$_B$-E2 interaction, we have compared the kinetics of WT Nse2 and Δ4C-Nse2 (ΔSIM2) in single-turnover SUMO conjugation on $_C$p53 substrate. The kinetic analysis displayed a 4-fold reduction of the Nse2 catalytic properties when SIM2 was removed, with $K_{cat}/K_M$ values around 0.025 and 0.007 for WT and ΔSIM2, respectively (Fig. 4e, left). Under these in vitro conditions, the absence of E3 did not retrieve measurable parameters. Interestingly, Nse2-ΔSIM2 reactions with extra WT or Smt3D68R yielded similar kinetic values (Fig. 4e, right), indicating that in the absence of SIM2, the extra SUMO in the reaction does not enhance the discharge reaction. These results show a connection between the SIM2 deletion (Δ4-Nse2) and the disruption of the SUMO$_B$ E2 interface (Smt3-D68R), shedding light on the role of SIM2 to anchor SUMO$_B$ to the E2 backside.

**SUMO$_B$ in the kinetics of the E2-SUMO discharge reaction**. A deeper kinetic analysis was conducted using purified E2~SUMO$_D$ thioester linked to a fluorophore (Alexafluor488) as substrate in single-turnover reactions, in which the only components are the E2~Smt3 thioester donor, the Smc5/Nse2 E3 ligase, and the $_C$p53 substrate acceptor. The results indicate that the presence of SUMO$_B$ increases by 3-fold the catalytic constant ($K_{cat}$) but yields similar affinity constants ($K_M$) (Fig. 5b, c). SUMO$_B$ interaction probably contributes to the "optimal" stabilization of the E2~SUMO thioester. Interestingly, the curves of initial velocities *vs* substrate could only be plotted to a sigmoidal equation (Hill equation), which is characteristic of cooperative behavior in multi-interfaced enzyme kinetics (Fig. 5b, c).

The role of ssDNA, which was reported to enhance the E3 activity of Nse2 by binding to the Smc5/Arm subunit[41], was also assessed in the kinetic analysis. Interestingly, the absence of ssDNA shifts the curve to the right (Fig. 5b), resulting in a 4-fold increase in the $K_M$, from 2.7 to 8.5 μM (Fig. 5c). Such shift in the sigmoidal curve indicates a positive cooperativity, confirming the stabilization and/or an E3 rearrangement upon ssDNA binding[41]. Globally, the kinetic analysis of the discharge reaction confirms that the SIM2-SUMO$_B$-E2 interaction increases the catalysis, and that ssDNA binding increases substrate affinity.

Finally, a fusion between C-terminal tail of Nse2 and SUMO$_B$ was engineered to investigate the influence of a constitutive SUMO$_B$ present at the E2 backside. Gel filtration chromatography showed an increased stability of the complex formed between the E2-SUMO mimetic and the fused Nse2-SUMO$_B$, compared to WT-Nse2, which did not elute as a single peak (Fig. 5d). Single-turnover discharge reactions with the E2~SUMO$_B$ fluorophore substrate displayed a remarkable rate increase with the Nse2-SUMO$_B$ fusion, highlighting the entropic benefit of increasing the local concentration of SUMO$_B$ at the E2 backside. Strikingly, a single D68R point mutant in the fused SUMO$_B$ decreased dramatically the rate constants (Fig. 5f, g), underlining the major role of the SUMO$_B$ E2 interface in the reaction. Additionally, restriction of SUMO$_B$ flexibility by shortening the N-terminal extension (Δ18-Smt3) also reduced the catalysis (Fig. 5f, g). Finally, the absence of ssDNA decreased the catalytic efficiency by 9-fold, and as observed in reactions with non-fused Nse2, by increasing the $K_M$ value. Interestingly, all kinetic curves with the

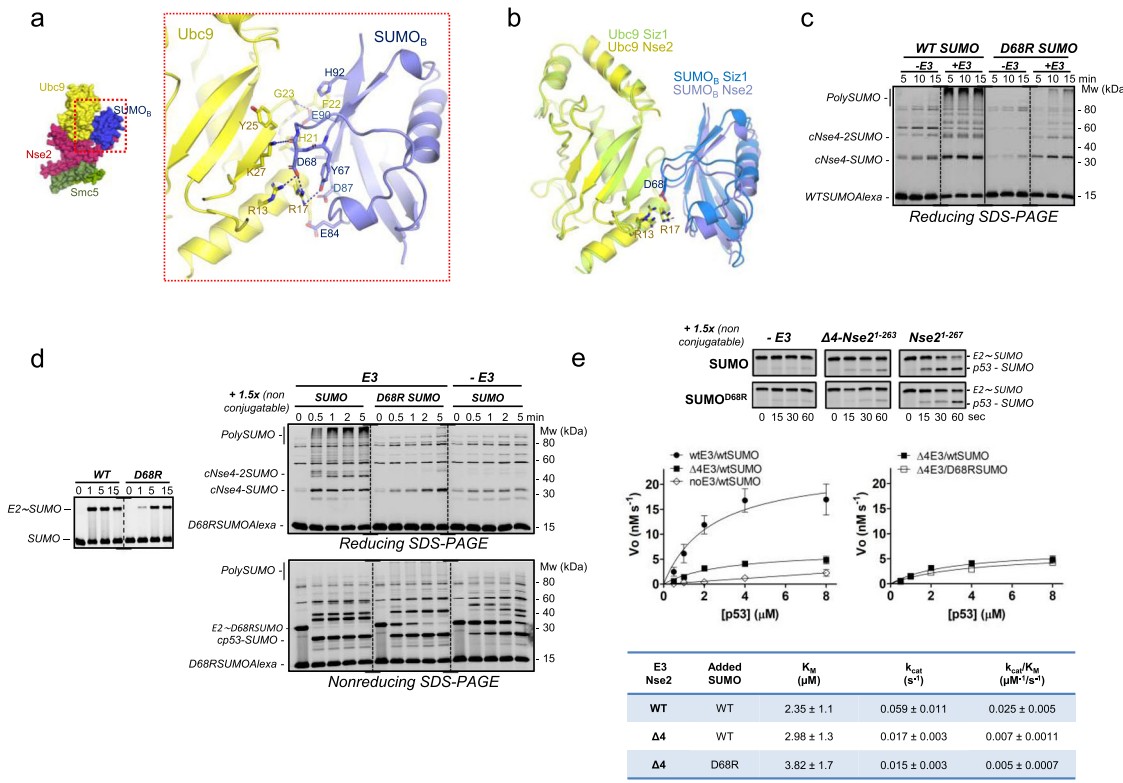

**Fig. 4 The SUMO$_B$ backside is fixed by the C-terminal SIM2 of Nse2. a** Cartoon representation of the interaction between Ubc9 and SUMO$_B$ (Asp68 SUMO makes electrostatic contacts with Ubc9's Arg13 and Arg17). **b** Structural alignment of Ubc9 in Nse2 and Siz1 complexes (5JNE), revealing a displacement of the backside SUMO$_B$. **c**. Multiple-turnover SUMOylation reactions using of wild type or D68R SUMO in the absence or presence of yeast Nse2/Arm-Smc5 E3 complex at 30 °C. **d** Single-turnover SUMOylation reactions stopped by EDTA using $^{D68R}$SUMO$_D$~E2 thioester by yeast Nse2/Arm-Smc5 E3 complex in the presence of 1.5-fold excess non-conjugatable SUMO (WT or D68R) at 30 °C. $^{D68R}$SUMO is activated by E1 and transferred to the E2 at lower rates as the wild type (left). $^{WT}$SUMO triggers the E2 discharge on $_c$Nse4 or $_c$p53 compared to $^{D68R}$SUMO. SDS-PAGE represents one of three technical replicates. **e** Kinetics of discharge reactions of $^{D68R}$SUMO$_D$~E2 thioester by WT-Nse2 and Δ4C-Nse2 (ΔSIM2) in the presence of non-conjugatable SUMO (WT or D68R) using increasing concentrations of $_c$p53 substrate, (0.5 to 8 μM) at 2 °C. Representative SDS-PAGE of reactions at 2 μM of $_c$p53. The quantified rate data show mean ± SD ($n$ = 3 technical replicates). Kinetics parameters obtained from the curves upon fitting using M-Menten equation in Prism (GraphPad) are shown (bottom). For gel source data, see Supplementary Fig. 3 and Source data file.

Nse2-SUMO$_B$ fusion fitted to a hyperbolic equation, in contrast to the sigmoidal curves for WT Nse2, perhaps indicating a loss of cooperativity when the SIM2-SUMO$_B$ interaction is excluded from the complex.

**Conformational change of Loop-SIM1 upon binding to the E2-SUMO$_D$ donor**. Our complex structure reveals a direct interaction of SUMO$_D$ with elements outside the SP-RING, namely the Loop SIM1, which undergoes an important conformational change (18 Å movement of Cα Asp169) (Figs. 1, 6a). Nse2 SIM1 adopts a SIM-like β conformation in contact with SUMO$_D$, forming an extended antiparallel β-strand with several main chain hydrogen bonds (Fig. 6b). Normally SIM motifs contain two hydrophobic residues buried in a SUMO cavity between α-helix and the edged β-strand. In yeast Nse2 SIM1, such positions are occupied by polar residues, Asp172 and Gln174, not optimal for a SIM binding. However, Loop SIM1 compensates the binding affinity by the presence of additional electrostatic interactions with SUMO$_D$ residues surrounding the SIM1 cavity. In particular, Glu170 and Asp171 are engaged with Arg47 and His23, respectively; and SUMO$_D$ Arg55 conducts polar interactions with the main chain oxygen of Ile161 and the side chain of Gln174 (Fig. 6b). Sequence alignment between Loop SIM1 in Nse2 orthologs reveals poor conservation (Fig. 6c), remarking the role of the main chain hydrogen bonding in the SUMO$_D$ interface. Only the acidic region bordering the SIM1 motif, which is

involved in electrostatic interactions with SUMO$_D$, displays some sequence conservation (Fig. 6c).

In vitro conjugation analysis with three Nse2 Loop SIM1 mutants: Nse2 ΔLoop1-SIM1 (deletion from Val160 to Gly176), Nse2 G177P point mutant, and Nse2 E170R/D171R/D172R triple mutant; showed in all instances a strong decrease in SUMO conjugation compared to WT-Nse2, up to 90% reduction for the ΔLoop1-SIM1 deletion mutant (Fig. 6d). These results underline the essential function of E3 interfaces outside the SP-RING to fix the E2~SUMO$_D$ thioester in an optimal catalytic orientation, in this case by direct contact of SIM1 with SUMO$_D$ after an important conformational rearrangement.

Also, to analyze the role of the SIM1 in vivo, we integrated mutations of the SIM1 at the endogenous *NSE2* locus. Therefore, we generated *nse2* mutants bearing a ΔLoop1-SIM1 deletion (*nse2-Δ160-176*), a substitution of three conserved negatively charged residues in this loop (Glu170, Asp171, and Asp172) to positively charged arginine residues (*nse2-EDD-RRR*) and a G177P mutation (*nse2-G177P*). As shown in Fig. 6e, all three mutants were sensitive to MMS, indicating that the SIM1 is important for the repair of alkylation damage in budding yeast.

**Extensive SP-RING interaction in Nse2 to bind the E2~SUMO$_D$ thioester**. Our complex structure shows an extensive interaction between the SP-RING domain and Ubc9-SUMO$_D$. Despite particular contacts, the Ubc9 interface

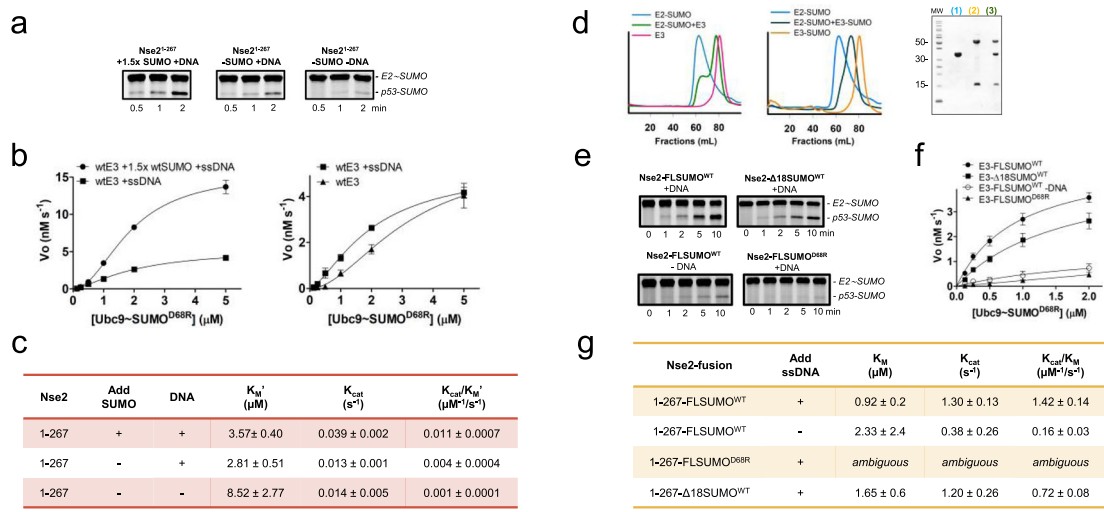

**Fig. 5 Effect of SUMO_B backside and ssDNA on E3 activity of Nse2-Arm/Smc5. a** Representative SDS–PAGE reactions at 1 μM Ubc9-SUMO$^{D68R}$–Alexa488 thioester. **b** Kinetics of single-turnover reactions using purified E2Ubc9-SUMO$^{D68R}$–Alexa488 thioester, ±1.5-fold excess of non-conjugatable SUMO$_B$, ±50nt ssDNA, $_c$p53, and non-fusion E3 (400 nM) at 30 °C. **c** Kinetics parameters were obtained using Hilll's sigmoidal equation in Prism (GraphPad). The quantified rate data show mean ± s.d. ($n = 3$ technical replicates). For gel source data, see Supplementary Fig. 4. **d** Gel filtration chromatography and SDS-PAGE of Nse2 and Nse2-SUMO fused in complex with Ubc9-SUMO$_D$ thioester mimetic. **e** Representative SDS–PAGE reactions at 1 μM E2Ubc9-SUMO$^{D68R}$–Alexa488 thioester. **f** Kinetics of single-turnover reactions using purified E2Ubc9-Smt3$^{D68R}$–Alexa488 thioester, ±50nt ssDNA, $_c$p53, and indicated fusion E3-SUMO (4 nM) at 30 °C. **g** Kinetic parameters were obtained using M-Menten's equation in Prism (GraphPad) are shown. The quantified rate data show mean ± SD ($n = 3$ technical replicates). For gel source data, see Supplementary Fig. 5 and Source Data file.

resembles the canonical binding of RING-like E3 ligases with E2 enzymes, either from the SUMO or ubiquitin family. In Ubc9 the interface includes hydrophobic and polar contacts from the α1-helix and two-loop/coil regions, such as Leu4, Gln7, and Arg8 from α1-helix; Pro69 and Ser70 from the β3-β4 loop; and Glu99, Arg104 and Pro105 from a coil after α2-helix (Fig. 7a). It is worth remarking the electrostatic bonds engaged between Ubc9's Glu99 and Arg104, with Arg219 and Asp220 from the SP-RING of Nse2, respectively. On the SP-RING side, two regions participate in the interface: the loop around Ile186, mostly conserved in E3 RING domains; and residues next to the $Zn^{2+}$ site, namely Tyr212, Arg219, and Asp220 (Fig. 7a, b).

In Nse2, in contrast to the contacts observed in Siz1, the SP-RING interacts with SUMO$_D$ through His202 ($Zn^{2+}$ ligand), forming two hydrogen bonds with main chain oxygens of Gly57 and Gln56; and through the loop from Gln223 to Ser227 with SUMO$_D$ surface residues from Lys58 to Ser62 (Fig. 7b and Supplementary Fig. 6). The different contacts engaged by the Nse2 SP-RING are mainly due to the insertion of two alanines (Ala224-Ala225) between the $Zn^{2+}$ ligands Cys221 and Cys226 (Fig. 7c and Supplementary Fig. 6), and by the hydrogen bonds formed by His202 ($Zn^{2+}$ ligand), not present in the Siz1 complex but also observed in ubiquitin RING domains[7,8,17]. Structural alignment between SP-RING domains (rmsd 1.89 Å, 20% identity) shows an extensive SUMO$_D$ interface area in Nse2, 234 Å2, compared to Siz1, 168 Å2 (Fig. 7c, d), which does not engage any hydrogen bond with SUMO$_D$. Globally, Nse2 and Siz1 have developed different interfaces to fix the donor SUMO$_D$ in a "closed" conformation for the E3-dependent discharge reaction, either outside the SP-RING domain (SIM1 and SIM2 interfaces) or by different contacts between SP-RING and the donor SUMO$_D$ (Figs. 1c, d, 2c, 6b, 7a, b, 8).

## Discussion

There are only a few bona fide E3 ligases in the SUMO pathway, one type shares similarities with the ubiquitin E3 ligases, containing a characteristic SP-RING-like domain, and the other two types, RanBP2 and ZNF451, are not related to any other type of

E3 ligase. However, all enhance SUMO conjugation by anchoring the E2~SUMO thioester in a closed or optimal conformation to form the isopeptidic bond on the substrate. In all three types of SUMO E3 ligases, the closed E2~SUMO conformation is anchored by contacts with the donor SUMO$_D$ utilizing SIM motifs located outside the SP-RING domain. This interaction was first described in RanBP2, and later reported in ZNF451 and Siz1[11,14,17], and in all cases, a SIM-like motif interacts with the same cavity in the SUMO surface. Ubiquitin RING E3 ligases also fix the E2~ubiquitin thioester in a closed conformation by direct contacts between the donor ubiquitin with the RING domain and by elements outside the RING domain, resembling the SIM-like motifs in SUMO. In dimeric ubiquitin RING E3 ligases, such as RNF4 or BIRC7, the donor ubiquitin binds both RING domains in the dimer[7,8]. In a monomeric RING, such as Cbl-B, a phosphorylated tyrosine in a tail before the RING domain contacts the donor ubiquitin[48]. In Ark-like E3 ligase RING domains, an additional ubiquitin binds the RING domain and enhances the discharge reaction[49]. In SUMO SP-RING E3 ligases, such as Nse2 or Siz1, the interface between the non-RING elements (SIM-like motif) and the donor SUMO$_D$ is more extensive than in ubiquitin RING E3 ligases. Interestingly, the SP-RING in Nse2 uses similar contacts as ubiquitin RING E3s (e.g., RNF4 or cIAP1) to bind SUMO$_D$, such as the hydrogen bonds between His202 ($Zn^{2+}$ ligand) and the donor SUMO$_D$, intriguingly such interaction was not observed in the SP-RING of Siz1 (Supplementary Fig. 6)[17].

Additionally, structures of primed SUMO E3 ligases complexes have revealed the presence of additional anchor points to fix the E2~SUMO$_D$ thioester, such as the binding of a second SUMO at the backside of the E2 enzyme. In the complex structure of the ZNF451 E3 ligase, a second SIM motif binds SUMO$_B$ at the Ubc9 backside and is essential for the E3 ligase activity[14]. In the Siz1 structure, SUMO$_B$ was also observed bound to the E2 backside. But in this instance, the binding was prompted by the use of the Siz1-SUMO$_B$ fusion, although biochemical data showed a role of a C-terminal SIM motif in Siz1 to anchor SUMO$_B$ at the E2 backside[17]. In contrast, the complex structure with the IR1-M fragment of RanBP2, tethering to the E2 backside seems to be

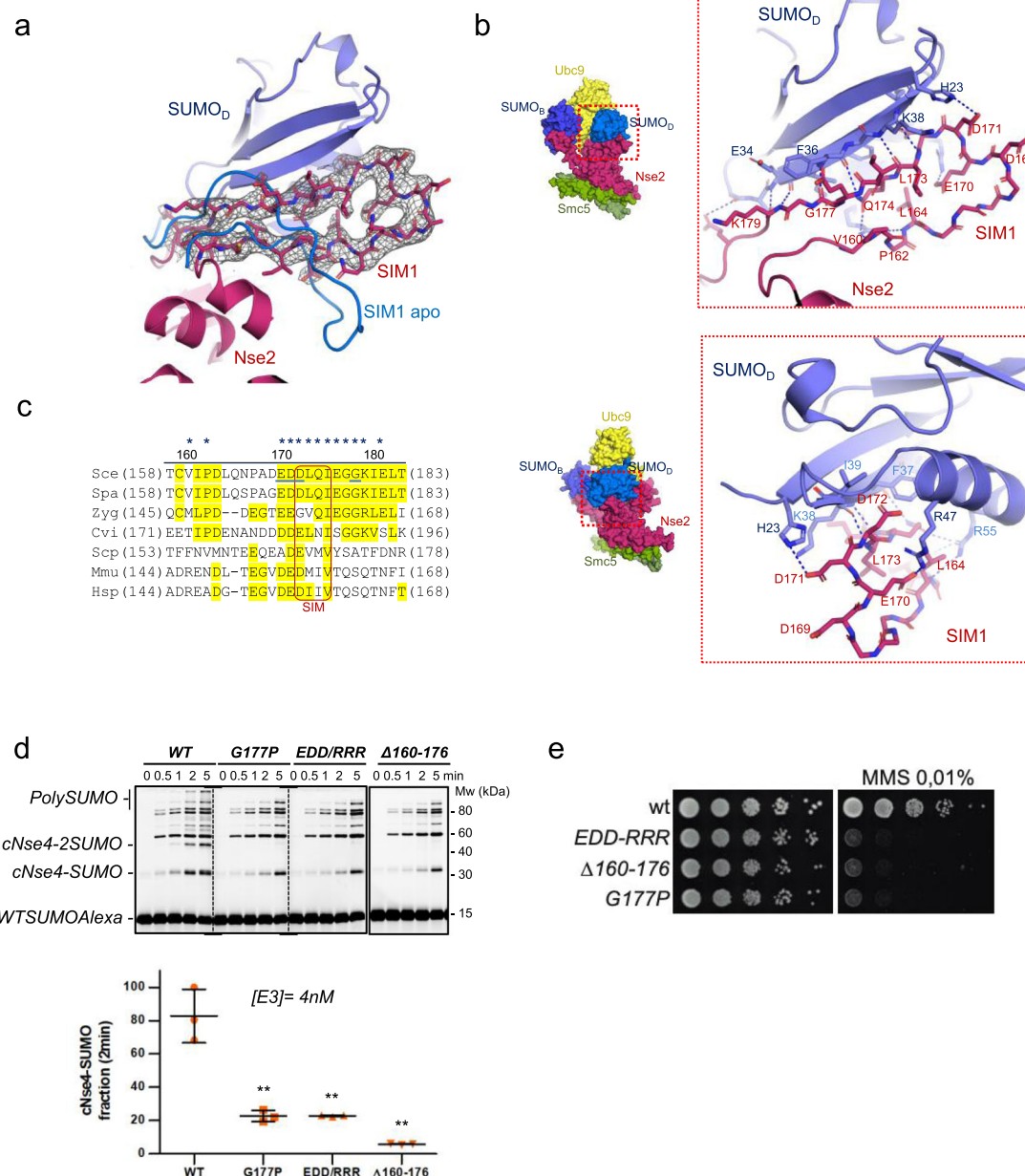

**Fig. 6 Conformational change of Loop-SIM1 upon binding to the E2-SUMO$_D$ donor. a** Structure and electron density maps of Loop SIM1. **b** Cartoon representation of the complex between Loop SIM1 Nse2 (V160 to K179) and SUMO$_D$. **c** Multiple alignment of SIM1 sequences from *Saccharomyces cerevisiae, S. eubayanus, S. paradoxus, Zygos. parabailli, Schizos. pombe, Mus musculus* and *Homo sapiens* (poor conservation except for the acidic region). **d** Multiple-turnover SUMOylation reactions of yeast Nse2-Arm/Smc5 complex (wild type and mutants) using cNse4 at 30 °C. Error bars showing that point mutations or deletion of SIM2 reduce E3 activity. Data values represent the mean ±SD, *n* = 3 technical replicates. Significance was measured by a two-tailed unpaired t-test relative to wild-type. All data were analyzed with a 95% confidence interval. **P < 0.005. Exact *P* values from the left to right: 0.0032, 0.0029, 0.0011. Source data are provided as a Source Data file. **e** Growth test analysis of wild type, *nse2-EDD-RRR, nse2-Δ160-176* and *nse2-G177P* cells; 10-fold serial dilutions of liquid cultures were spotted in YPD in the presence or not of MMS 0.01% and incubated at 30 °C for 60 h.

conducted directly by a short α-helix of the RanBP2 E3 ligase, which binds Ubc9 in a similar location as the backside SUMO$_B$[11]. In all three cases, the E3 ligases seem to embrace and stabilize the charged E2~SUMO thioester, in Siz1 and ZNF451 through a SIM-SUMO$_B$ interaction. In our primed complex structure with the Nse2 E3 ligase, the electron density maps clearly show a direct interaction of the C-terminal SIM motif of Nse2 (SIM2) at the backside SUMO$_B$ (see model in Fig. 8), a mechanism probably shared by other SP-RING E3 ligases (PIAS or Siz1)[46]. SIM motifs can be arranged parallel or antiparallel with SUMO, involving

different side-chain interactions, either hydrophobic, buried in the SUMO cavity, or electrostatic, between negatively charged SIM residues with SUMO positive residues[11,15,50–52]. Interestingly, our SIM2-SUMO$_B$ structure shows two conserved hydrophobic residues buried in the SUMO$_B$ cavity and stands out the electrostatic interaction between the SIM C-terminal carboxylate and Arg47 of SUMO$_B$, all contribute to increasing the binding affinity. The SIM2-SUMO$_B$ interaction in Nse2 resembles the structures of SUMO with SIM peptides[53–55], such as the structure of SUMO1 bound to a peptide of the C-terminal human PIAS1

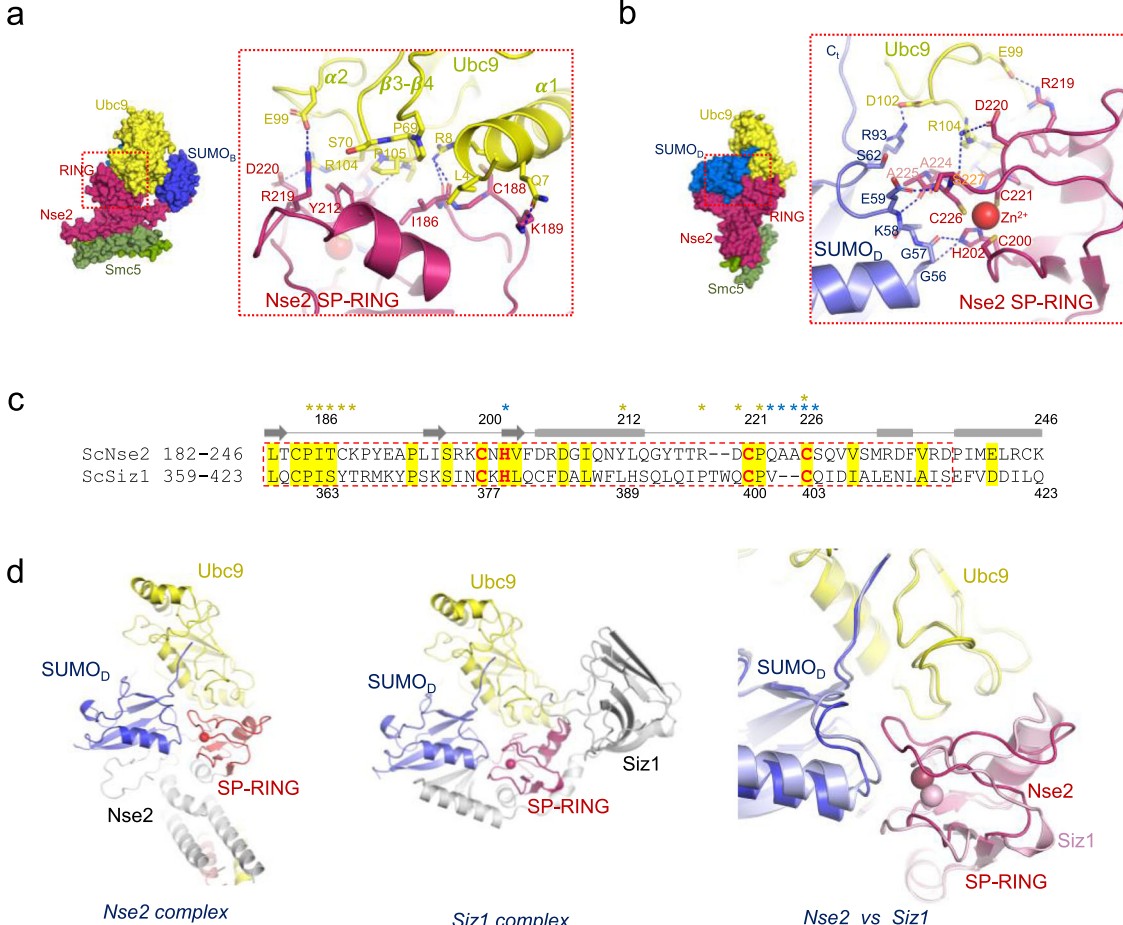

**Fig. 7 Extensive SP-RING interaction with the E2-SUMO_D thioester mimetic. a** Cartoon representation of the complex between SP-RING Nse2 (Ile186 to Ser227) and Ubc9. **b** Cartoon representation of the complex between SP-RING Nse2 (Ile186 to Ser227) and SUMO_D (Gly53 to Arg93). **c** Sequence alignment of SP-RING domains of yeast Nse2 and Siz1. Asterisks indicate residues that make interactions with Ubc9 (yellow) and SUMO_D (blue). Secondary structure elements are depicted above (middle). **d** Structural alignment comparison of SP-RING domains in Nse2 and Siz1 complexes (5JNE).

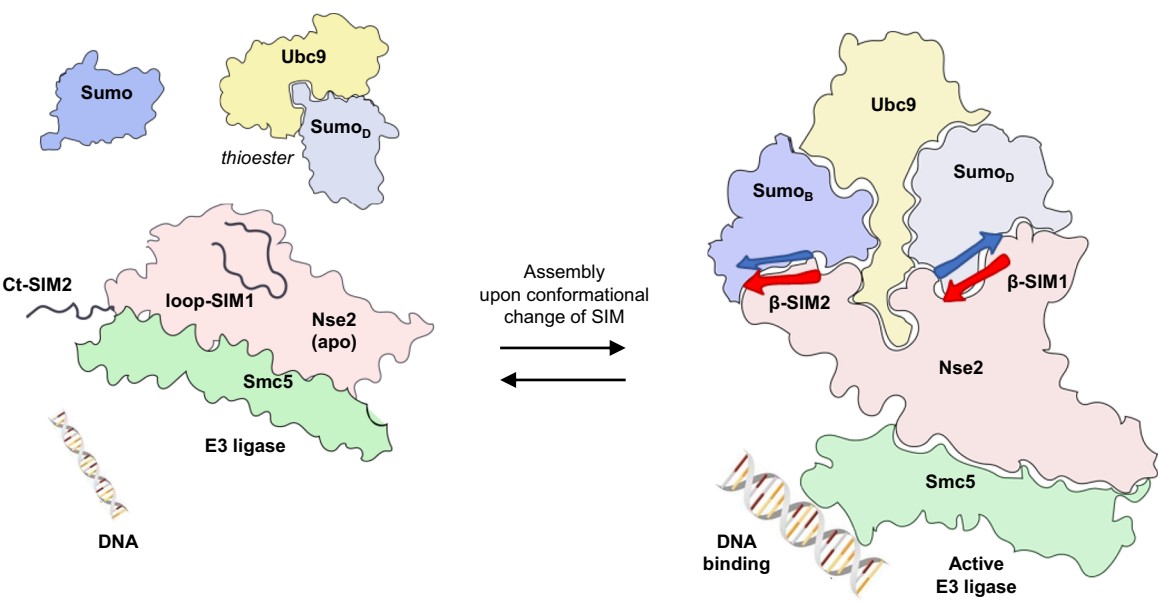

**Fig. 8 Model of Arm-Smc5/Nse2 interaction with E2-SUMO_D thioester and SUMO_B backside.** DNA associates with Smc5 leading to conformational changes of Nse2 SIM1 and SIM2 which clamps E2-SUMO thioester and SUMO backside into the closed and active conformation.

(Fig. 2d)[46], displaying a similar location for the hydrophobic side chains of positions 1 and 3 of the SIM motif, but with a different C-terminal carboxylate engagement with a SUMO arginine.

For some ubiquitin RING E3 ligases, the backside ubiquitin-E2 interaction has been reported to enhance the E3 discharge reaction by an allosteric mechanism, in which structural changes in the α1-helix and α1-β1 loop of E2 stabilizes the RING E3-E2~ubiquitin complex[31,56]. In our primed Nse2 structure, direct contact between the E2 α1-β1 loop and the donor $SUMO_D$ is not observed, as seen between donor ubiquitin and E2[31,56] and the structure of the E2 α1-β1 loop does not show any evident distinct conformation in the complex. However, our kinetic analyses using the E2~SUMO thioester substrate suggest a cooperative mechanism of the discharge reaction, which could only be fitted to a sigmoidal equation (Fig. 5a). Interestingly, such cooperativity is lost when the SIM2 interaction is not present, as occurs with the Nse2-$SUMO_B$ fusion (Fig. 5b). It would be interesting in the future to explore whether this cooperative behavior observed in the kinetics of the discharge reaction, in which SIM2 plays a relevant role, has a mechanistic explanation in terms of structural remodeling.

The kinetic analysis also sheds some light on the enhancement of the E3 activity of Nse2 by binding to DNA through the Smc5 Arm subunit[41]. The kinetic parameters displayed around a 4-fold increase of the $K_M$ in the absence of ssDNA, either for the fused Nse2-$SUMO_B$ or for the WT-Nse2, suggesting a role for ssDNA as a positive allosteric factor in the E3-catalyzed discharge reaction. It has been well established the connection between the activation of the Smc5/6 complex by the ATPase activity of the head domains and the increase of affinity for DNA, where it fulfills its biological function during DNA repair processes[37,38]. In such Smc5-Smc6-DNA-bound arrangement, a DNA-dependent activation of the Nse2 SUMO E3 ligase activity would restrict SUMO conjugation to protein substrates in the vicinity of the DNA repair sites[37,38,41], avoiding non desired SUMO conjugation by the non-substrate specific activity of the Nse2 E3 ligase.

Recent low-resolution cryoEM structures and MS analysis of the Smc5/6 complex revealed the presence of different structural conformations, standing out the rod shape formed by the interaction of the two Arm domains of Smc5 and Smc6, in contrast to the O-ring shape, probably indicative of the active DNA-binding conformation of the complex[37,38,57]. In such scenario, other possible regulatory mechanisms for the SUMO E3 ligase activity of Nse2 could be envisaged, in which other subunits within the complex might communicate with Nse2 to regulate the E3 activity[42]. Yeast two hybrid experiments have shown that the Nse5 subunit in the Smc5/6 complex interacts with several members of the SUMO pathway, including the PIAS family Siz1 and Siz2 E3 SUMO ligases and the E2-conjugating enzyme, Ubc9[58–60]. These interactions suggest that Nse5 may promote Nse2-dependent sumoylation, a possibility that is supported by the observation that *nse5* hypomorphic mutants reduce sumoylation of Nse2 targets[42,60]. It is also interesting to note that the DNA damage checkpoint kinase ATR can phosphorylate the C-terminal domain of Nse2 to promote its SUMO-ligase activity. Phosphorylation occurs in two serine residues (Ser260 and Ser261) located next to SIM2[61], probably increasing the affinity for positive-charged residues on the SUMO surface. Despite our in vitro SUMO reactions with phosphomimetic S260E, S261E and S260E/S261E Nse2 constructs do not display a substantial SUMO conjugation increase compared to WT (Supplementary Fig. 7), we cannot discard that SIM2 may be a Phospho-SIM in vivo, further enhancing SUMO ligase activity under conditions that activate the DNA damage checkpoint. This situation might be particularly important under conditions of higher level of endogenous DNA damage, such as those involving inactivation of genes involved in DNA repair (Fig. 3).

Interestingly, it has been recently reported the interaction between SIM-like motifs present in the Nse5 subunit with Nse2, which mighty regulate the SUMO E3 ligase activity perhaps by mechanisms similar to ones described in the present paper[39]. In a different scenario, a possible regulatory mechanism within the Smc5/6 complex might involve binding of SUMOylated subunits of the Smc5/6 complex to the E2 backside of the E2~$SUMO_D$ thioester, resembling the $SUMO_B$ backside interaction observed in the present structure. This mechanism was already envisioned for the ZNF451 E3 ligase, in which SUMO conjugation in distant regions of ZNF451 might provide the $SUMO_B$ backside necessary to fulfill its E3 ligase activity[12,14,18]. In yeast, SUMOylation of the Smc5 subunit has been shown to be involved in the error-free bypass of DNA lesions[47]. Perhaps this mechanism is more relevant in humans, in which the lack of an evident SIM2 at the Nse2 C-terminus might favor the binding at the E2 backside of other SUMOylated subunits within the Smc5/6 complex. It would be very exciting in the future to identify novel regulatory mechanisms of the SUMO E3 ligase activity of Nse2 within the context of the full Smc5/6 complex.

## Methods

**Plasmids construction for overexpression in bacterial cells.** Full-length SMC5 from *Saccharomyces cerevisiae* S288c cloned into pET28a vector between NcoI and XhoI were used to generated two constructs by PCR amplification, named Arm/SMC5 (Asp302-Thr366-Gly-Thr-Arg737-Gln813) and $_{short}$Arm/SMC5 (Asp328-Thr366-Gly-Thr-Arg737-Gln777) which were then individually cloned into pET28a vector. The full-length budding yeast NSE2 (MMS21) was cloned in pET15b between XhoI and BamHI (Nse2 1-267) and used to engineer a short construct ($_{short}$Nse2, ΔN26/Δ83-134). FLNSE21-267 was also used to construct three fusion versions where a SMT3 cDNA was inserted on the C-terminal of NSE2 without stop codon (FLNSE2-WTFLSMT3ΔGG, FLNSE2-WTΔ18SMT3ΔGG and, FLNSE2-D68RFLSMT3ΔGG). Mutagenesis protocol on FLNSE21-267 was used to prepare NSE2 mutant vectors: CtΔ16, CtΔ8, CtΔ4, I264P, V266R, I264A/V266A, G177P, E170R/D171R/D172R, Δ160-176, S260E, S261E. A list of the used primers can be found in Supplementary Table 2.

Cloning of cNse4 (c-terminal kleisin domain, Ile246-Asp402) or cp53 (c-terminal p53 domain, Lys320-Asp393) substrates, yeast E1 (AOS1:UBA2ΔC-term1-554), E2 (UBC9, WT, K153R, and A129/K153R), and precursor SUMO (SMT3: ΔN18, K11C, and K11C/D68R), have been described in detail[41,43,62].

**Protein expression and purification.** All Nse2 recombinant proteins containing N-terminal 6xHis-tag were co-expressed with Arm/Smc5 without tag in *Escherichia coli* Rosetta 2(DE3) cells (Novagen). Bacterial cultures were grown at 37 °C to $OD_{600} = 0.6$, before 0.5 mM IPTG addition. Cultures were then incubated for 16 h at 20 °C and harvested by centrifugation. Cell pellets were equilibrated in Lysis Buffer (20% sucrose, 50 mM Tris pH 8.0, 1 mM BME, 350 mM NaCl, 20 mM imidazole, 0.1% IGEPAL), and cells were disrupted by sonication. Cell debris was removed by centrifugation (22,000 g for 15 min). 6xHisNse2-Arm/Smc5 complexes were purified by metal affinity chromatography using Chelating Sepharose Fast Flow resin (GE Healthcare) and eluted with 20 mM Tris pH 8.0, 350 mM NaCl, 1 mM BME, and 250 mM imidazole. Fractions Nse2-Arm/Smc5 (FL, short, fusion, and mutants) were further purified by gel filtration column (Superdex 200 HiLoad; GE Healthcare) followed by ion-exchange chromatography equilibrated in 20 mM Tris pH 8.0, 50 mM NaCl, 1 mM BME (Resource Q; GE Healthcare). Protein complexes were eluted by 50-500 mM NaCl gradient.

Expression and purification of 6xHis-tag cNse4 (Ile246-Asp402) or cp53 (Lys320-Asp393) substrates, yeast E1 (Aos1:Uba2ΔC-term1-554), yeast E2 (Ubc9, WT, K153R, and A129K/K153R), and precursor SUMO (Smt3: ΔN18, K11C, and K11C/D68R), were individually expressed and purified following the aforementioned protocol. SENP2 protease was added to the fractions of Smt3 eluted from metal affinity chromatography to generate mature and conjugatable SUMO.

$^{A129K/K153R}$Ubc9-$^{FL}$Smt3 thioester mimetic was prepared according to Streich and Lima[17] in a buffer containing 20 mM BIS-TRIS propane (pH 9.5), 50 mM NaCl, 10 mM $MgCl_2$, 0.1% Tween-20, 2 mM ATP, 0.5 µM E1, 100 µM Ubc9A129K/K153R, and 200 µM Smt3 for 1 h at 30 °C and purified by Superdex75 equilibrated in 20 mM Tris pH 8.0, 350 mM NaCl and 1 mM BME. Purified sample was dialyzed to decrease salt concentration up to 10 mM. Finally, sample was additionally purified by Resource Q equilibrate in the same buffer without salt, Ubc9-thioester mimetic was eluted using a 0-400 mM NaCl gradient.

**Crystallization, data collection, and structure determination**. For crystallization of the $_{short}$Nse2/Arm-Smc5 E2-SUMO$_D$ SUMO$_B$ complex, purified $_{short}$Nse2/Arm-Smc5, $^{ΔN18}$Smt3, and $^{A129K/K153R}$Ubc9-$^{FL}$Smt3 thioester mimetic were mixed in equimolar concentrations (~7 µM of each protein) in 20 mM Tris-HCl (pH 8.0) containing 100 mM NaCl and 1 mM BME. The complex was concentrated up to 9 mg/mL (120 µM) and mixed 1:1 ratio with 12% PEG 8000, 0.1 M MES pH 6.5, 0.2 M dimethyl-2-hydroxyethylammoniumpropane sulfonate (NDSB 211), 8% ethylene glycol. Crystals appeared after 3 days using the hanging-drop vapor diffusion method at 18 °C. Crystals were cryo-protected in a reservoir buffer containing 20% ethylene glycol and flash frozen in liquid nitrogen prior to diffraction analysis.

Diffraction data were recorded from cryo-cooled crystals (100 K) at the ALBA synchrotron in Barcelona (BL13-XALOC beamline)[63]. Data were integrated and merged using XDS[64] and scaled, reduced, and further analyzed using CCP4[65]. The crystal grown at the buffer system belongs to the P2$_1$2$_1$2$_1$ ($_{short}$Nse2/Arm-Smc5 E2-SUMO$_D$ SUMO$_B$ complex) space groups. Phasing and model-building were obtained by molecular replacement using Phaser-MR from PHENIX[66]. Refinements and model rebuilding were performed using PHENIX[66] and Coot[67]. Figures were generated using PyMOL (http://www.pymol.org). The data-collection and refinement statistics are summarized in Supplementary Table 1.

**Smt3 labeling with Alexa Fluor488**. Mature $^{K11C}$Smt3 (wild type and D68R) were fluorescent labeled using Alexa Fluor488 Maleimide C5 according to the manufacturer's instructions (Invitrogen). Protein was diluted in 20 mM HEPES pH 7.2, 50 mM NaCl, 0.5 mM TCEP up to 40 µM. Alexa Fluor488 stock solution was added gently to reach a 20:1 ratio. Mixtures were kept at 4 °C by 16 h. Free probe molecules were removed using PD-10 desalting column (Cytiva), followed by 10x times volume washing by Centricon (MerckMillipore) centrifugation equilibrate on the same HEPES buffer. Proteins were concentrated up to 2 mg/mL and flash frozen prior use.

**Multiple-turnover SUMOylation assay**. Reactions were performed as described by Varejão et al.[41] with some modifications. 25 µL of a reaction mix containing 40 mM HEPES pH 7.5, 10 mM MgCl$_2$, 0.2% Tween-20, 25 mM NaCl, 4 mM dithiothreitol, 0.8 µM of ssDNA 50nt, 2 µM mature Smt3$^{K11C}$-Alexa488 (wild type or D68R), 6 µM cNse4 or cp53, 0.3 µM yeast E1 (Aos1:Uba2$^{ΔC-term1–554}$), 0.2 µM yeast E2 ($^{WT}$Ubc9) were incubated with 25 µL of filtered purified water plus 4 or 400 nM yeast E3 ($^{1-267}$Nse2/Arm-Smc5$^{309–815}$: wild type, Δ16, Δ8, Δ4, I264P, V266R, I264A/V266A, G117P, E170R/D171R/D172R, Δ160-176). The reaction was immediately initiated by the addition of 2 mM ATP and conducted at 30 °C. Samples were taken at the indicated time points and stopped with a 4X Laemmli sample buffer with or without BME (BioRad). Reactions using human E1 (Sae1:Sae2), E2 (Ubc9) and mature SUMO1$^{S9C/C52A}$-Alexa488 were also carried out under similar conditions.

**Single-turnover assay using E2-thioester formation stopped by EDTA**. Reactions were performed as described elsewhere[14,68] with minor changes. The E2-thioester was formed in a reaction mix that includes 20 mM HEPES (pH 7.5), 50 mM NaCl, 10 mM MgCl$_2$, 0.1% Tween-20, 0.1 mM dithiothreitol, 0.2 µM yeast E1, 2 µM yeast E2 (Ubc9$^{K153R}$), and 2 µM Smt3$^{K11C/D68R}$-Alexa488. The reaction was initiated by the addition of 0.5 mM ATP and was incubated at 30 °C for up to 5 min. Then, the reaction was quenched with the addition of 5 mM EDTA. To follow the thioester transfer mediated by the E3, 25 µL of formed Ubc9$^{K153R}$~Smt3$^{K11C/D68R}$-Alexa488 thioester solution were diluted with 25 µL water, and supplemented with, 0.8 µM ssDNA 50nt, 1.5 µM extra non-labeled Smt3 (wild type or D68R), 8 µM of cNse4 or cp53, and 400 nM wild type E3 ($^{1-267}$Nse2/Arm-Smc5$^{309–815}$). Reactions were incubated at 30 °C and samples were taken at indicated time points after the addition of the E3.

Additionally, to follow the kinetics of thioester discharge induced by the E3 (wild type and Δ4), 15 µL of formed Ubc9$^{K153R}$~Smt3$^{K11C/D68R}$-Alexa488 thioester solution were diluted with 15 µL water, and supplemented with 0.8 µM ssDNA 50nt, 1.5 µM extra non-labeled Smt3 (wild type or D68R) in the presence of increasing concentrations of cp53 (0-8 µM), and 400 nM wild type or Δ4 E3 ($^{1-267}$ or $^{1-263}$Nse2/Arm-Smc5$^{309–815}$). Reactions were performed on ice-bath (2 °C). Samples were taken and mixed with a 4X Laemmli non-reducing sample buffer (BioRad). Curves were fitted with the Michaelis-Menten equation using Prism (GraphPad).

**Ubc9$^{K153R}$-Smt3$^{K11C/D68R}$-Alexa488 thioester purification**. Ubc9$^{K153R}$-Smt3$^{K11C/D68R}$-Alexa488 thioester was formed and purified as described by Streich and Lima[17]. Briefly, the reaction mixture contained 20 mM HEPES (pH 7.5), 50 mM NaCl, 10 mM MgCl2, 0.1% Tween-20, 2 mM ATP, 0.4 mM DTT, 11 µM E1, 220 µM Ubc9$^{K153R}$, and 100 µM Smt3$^{K11C/D68R}$-Alexa488 was incubated for 5 min at 30 °C and purified by Superdex75 equilibrated in 50 mM NaCitrate pH 5.5, 200 mM NaCl, 5% glycerol, concentrated up to 30 µM and stored at −80 °C before use.

**Kinetic curves using purified Ubc9$^{K153R}$-Smt3$^{K11C/D68R}$-Alexa488 thioester**. To follow the kinetics of thioester discharge mediated by Nse2/Arm-Smc5

constructs (non-fusion and Smt3-fusioned $^{WT}$E3), the purified Ubc9$^{K153R}$-Smt3$^{K11C/D68R}$-Alexa488 thioester was serial diluted in 20 mM NaCitrate pH 5.5, 100 mM NaCl, and 5% glycerol. Then, 10 µL of diluted thioester (ranging from 0.625–25 µM) were incubated 40 µL reaction mixture containing 40 mM HEPES pH 7.5, 25 mM NaCl, 0.1% Tween-20, 4 or 400 nM E3, and 32 µM cp53, 0.8 µM ssDNA 50nt. In indicated reactions, 1.5-fold excess of non-conjugatable Smt3 over thioester was added to the mixture. Also, ssDNA was removed in some experiments, as indicated. Reactions were incubated at 30 °C and samples were taken and quenched with a 4X Laemmli non-reducing sample buffer (BioRad). Curves were fitted with Allosteric-Sigmoidal equation using Prism (GraphPad).

**Analysis and quantification of SUMOylation products on SDS-PAGE**. All products of the time-course reactions were stopped with 4xLaemmli sample loading buffer (Bio-Rad), resolved by 12% SDS–PAGE (in reducing or non-reducing conditions as indicated in the figures legends) and visualized by Alexa488 fluorescence emission in a Molecular Imager Versadoc MP4000 System (Bio-Rad). Band densitometry was calculated by Quantity One 1-D (Bio-Rad).

**Intrinsic fluorescence**. Tryptophan emission spectra were obtained by setting the excitation wavelength at 295 nm and collecting emission in the 310–400 nm range using a Jasco FP-8200 spectrofluorometer. Arm-Smc5/Nse2 (wild type and CΔ16) were diluted to achieve 1 µM in 20 mM HEPES pH 7.5 containing 50 mM NaCl. After recording the emission spectra 0.2 µM ssDNA (50nt) was directly added to the cuvette and a new spectrum was recorded. The temperature was maintained at 30 °C.

**Construction of yeast mutant strains**. All yeast strains constructed in the BY4741 genetic background. The nse2-SIM2Δ allele was generated by chromosomal integration of a PCR amplified selection marker using primers designed to insert a STOP codon inserted before the last 4 residues, thus preventing translation of the SIM2 motif. The PCR was transformed into BY5563 and verified by PCR and sequencing. Double mutants in combination with mms4Δ, esc2Δ, slx4Δ and Smc5/6 thermosensitive alleles were obtained by crossing and sporulation. The nse2-EDD-RRR and nse2-G177P mutants were created by PCR using plasmid pNC2279 as template[42] and primers with tails containing the desired mutations. The purified PCR products, with homologous 5′ and 3′ ends, were transformed in MC1061 for recombinational cloning. The nse2-Δ160-176 allele was generated by PCR using plasmid pTR2395 as template and primers containing the deletion. The PCR product was phosphorylated, ligated, and transformed into DH5α. The mutagenized sequences were then fused to a selection marker by PCR and transformed into yeast. All integrations were confirmed by PCR and sequencing.

**Reporting summary**. Further information on research design is available in the Nature Research Reporting Summary linked to this article.

## Data availability
Structure reported has been deposited in the Protein Data Bank under accession code 7P47. All other data supporting the findings of this study are available within the article and its supplementary information files. Source data are provided with this paper.

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

## Acknowledgements

This work was supported by grants from the "*Ministerio de Ciencia, Innovación y Universidades*" PGC2018-098423-B-I00 to DR and PGC2018-097796-B-I00 to JT-R and by grant 2017-SGR-569 from "*AGAUR-Generalitat de Catalunya*" to JT-R.

DR acknowledges support from the Serra Hunter program from Generalitat de Catalunya. X-ray experiments were performed at BL-13 Xaloc beamline at ALBA Synchrotron with the collaboration of ALBA staff.

## Author contributions

N.V., J.L., H.B.-G. and D.R. conducted the crystallization experiments and all in vitro reactions. J.C.-F., G.B. and J.T.-R. conducted the in vivo yeast experiments. N.V., J.T.-R. and D.R. contributed to the correction and writing of the manuscript.

## Competing interests

The authors declare no competing interests.
