## [Peer Review File · Nature Communications]

Structural basis for the E3 ligase activity enhancement of yeast Nse2 by SUMO-Interacting Motifs

Editorial Note: Parts of this Peer Review File have been redacted as indicated to remove third party material where no permission to publish were obtainedREVIEWER COMMENTS

Reviewer #1 (Remarks to the Author):

To catalyze the SUMO discharge from the E2 to the substrate, bona fide SUMO E3 ligases orient the donor SUMO via a SUMO interaction motif (SIM) in a closed conformation. A second SIM that binds a scaffold SUMO in the backside of Ubc9 stabilizes the E2-E3 complex. This scaffold SUMO is essential for the ZNF451 family and improves Siz/Pias (SP) ligase activity. However, the SIMs mapped for the Siz1 are not conserved in Nse2/Mms21, raising the question how these enzymes execute E3 ligase function. The study proposed by Nathalia Varejao and colleagues addressed this important topic by solving the crystal structure of Nse2-Smc5 in complex with a SUMO charged E2 mimetic and one additional SUMO. Structural, biochemical and cellular analysis disclosed that also in this case two SIMs have key functions for Nse2 E3 ligase activity. Analog to the other SUMO E3 ligases, one SIM positions the donor SUMO, while the second SIM stabilizes the E2/E3 complex by sandwiching a scaffold SUMO on the backside of the E2. Furthermore, the role of ssDNA that was earlier shown to improve E3 ligase function was further analyzed. Provided data indicate that ssDNA rather increases the K_m than the K_{cat} . Overall, the proposed work provides important insights and should be published in a journal like Nature Communications, but needs revision for publication.

Major comments:

1. Figure 1B. The authors compare the activities of the Smc5/Nse2short with Smc5/Nse2 WT and describe that their activities are comparable and substantially not affected. Although, Figure 1B shows that Smc5/Nse2short is active, it is significantly reduced compared to WT. This needs rephrasing.
2. Figure 2 E-G. To draw clear conclusions about enzyme functions it is important to discriminate between substrate modification, Smt3-chains and modification of the enzymes in the in vitro reactions. All SUMO enzymes are efficiently modified in in vitro assays, especially E3 ligases are highly autosumoylated and assemble free or attached SUMO chains. This reviewer is aware that enzymes and substrates in this study share the same tag and likely for the proteins domains used as model substrates no specific antibodies are available. Preferable, one substrate specific analysis should be included. If this is not feasible, at least minus substrate controls have to be included.
3. All substrate sumoylation assays presented show that substrates are modified with multiple Smt3s. Is it understood, whether these model substrates are modified at multiple sites, with Smt3-chains or a combination of both? To these reviewer's understanding, cp53 has only one SUMO site but forms tetramers. The E2~Smt3 thioester is indicated in Figure 2E, suggesting a multiturnover reaction under non-reducing conditions was performed. This raises the question whether all bands represent conjugates or also protein-complexes are detected? In case this assumption is correct, samples separated upon reducing conditions would clarify this issue.
4. It is noticeable that the work of the Pichler lab is explicitly not cited. Is there a reason for this? The Pichler lab has identified and characterized the two SIM dependent mechanism for the SUMO E3 ligase ZNF451 (Eisenhardt et al, 2015, Koidl et al 2016) and proposed that a similar mechanism likely applies

for SP-RING E3 ligases (Pichler et al, 2017). This idea was even adapted in a recent review from the Reverter lab (Varejao, 2018), seemingly they are aware of these studies. The Pichler lab has further shown that the backside of Ubc9 contributes to SUMO chain assembly involving yeast Ubc9 sumoylation or a scaffold SUMO bound by ZNF451 (Klug et al, 2013, Eisenhardt et al, 2015, Pichler et al, 2017).

As Figure 2 demonstrates that poly-Smt3 conjugates drastically disappear in various Nse2 mutations targeting SIM2 but less severely by Nse2 mutations targeting SIM1 or the Nse-E2 binding interface (Figure 6) raises the intriguing question whether SP-RING E3 ligases also assemble SUMO chains via the E2 backside. Bands marked in Figure 2 as Poly-Smt3 can, as pointed out above, present substrate and enzyme modifications, attached and free SUMO chains. Many of these bands disappear even after 5 minutes reaction time in the presence of Nse2delta4 and a substrate (cNse4 and cp53) but to a lesser extent when automodification/coil is tested. These data suggest that some fraction of the high molecular smear represents E3 automodification and the bands that remain unchanged, likely present E1 (Aos1 and Uba2) and E2 modifications. E3 automodification is a cis-reaction and appears to depend less on SIM2, compared to trans-reactions required to modify the substrate.

Further support for a function of the E2-backside SUMO-interaction in chain formation is indicated in Figure 4 D demonstrating that c-Nse4-2Smt3 and cp53-2Smt3 conjugates disappear in the presence of 1.5 x non-conjugateable Smt3 D68R, while at longer time points even in the reactions lacking the E3 (later time points) cp53-2Smt3 was detected.

5. E2 discharge reactions in Figure 4: In the Figure legend it is stated “D68RSUMO is activated by E1 and transferred to the E2 in similar rate as the wild type (left).” The figure shown presents a significant reduced charging of the E2 with SUMO D68R! This needs rephrasing. However, in the discharge assays it appears that similar amounts of charged E2s were used. The discharge assays to cp53 but not to cNse4 detects the charged E2 enzyme to follow its discharge. Is the cNse4 gel separated under reducing conditions or why is the charged E2 not detected in this case?

6. E2 discharge reactions in Figure 5 display very weak discharge compared to Figure 4.

What is the reason for that? According to the understanding of this reviewer the main difference in the reactions is the use of Smt3-E3 fusions in Figure 5 compared to addition of free Smt3 in Figure 4. Considering the corresponding raw data shown in Figures S4 and S5, indicate that reactions containing free Smt3 allow poly-SMT3 conjugates that are not detected in any reaction using the Smt3-E3 fusion. This raises the question, whether the nature of these poly-Smt3 conjugates is E3 automodification that is blocked upon Smt3-E3 fusion (see also below)? Polysumoylation of cp53 is unlikely as it has only one sumoylation site and chain formation should be largely impaired by using Smt3D68R.

7. Substrate E3 interaction. To this reviewer's understanding, cp53 is a model substrate that is unlikely to directly interact with Smc5/Nse2short. The cp53 fragment displays the tetramerization domain and one sumoylation site but lacks the DNA binding domain. As the presented data indicate that ssDNA rather induces the K_m than the K_{cat} , this raises the question how cp53 interacts with the E2-E3 ligase complex. A low affinity interaction between cp53's SUMO consensus site and the catalytic cleft of the E2 is likely (Bernier-Villamor V et al, 2002) and sufficient for modification by the E2-E3 complex (Figure 2 E, left panel), especially at the high substrate concentrations used. As cp53 lacks its established DNA binding domain the question that arises is whether cp53 is able to interact with DNA. Alternatively, Figure S4 suggests that ssDNA reduces E3 automodification to the benefit of substrate modification? In the

absence of ssDNA, poly-SMT3 appears to increase compared to the ssDNA containing reactions. Is it possible, that E3 automodification competes with substrate modification and the E3 bound to ssDNA reduces automodification to the benefit of substrate modification? Titration of ssDNA to the reactions and discrimination between substrate and E3 sumoylation would clarify this issue.

Minor comments:

1. Smt3 and SUMO are used to describe the yeast SUMO, Smt3. For clarification, a uniform labeling would avoid confusion: one could either use Smt3 throughout the manuscript or probably γ SUMO or S.c.SUMO.
2. All gel presentations require size markers.
3. For clarification, reducing and non-reducing gels need to be indicated throughout the manuscript. It would also be helpful for the reader to state all enzyme concentrations in each Figure.
4. Ubc9-K153R is used for structural analysis and E2 discharge experiments, likely to prevent E2 sumoylation. However, the yeast Ubc9 is modified at K153 and K157 (Ho et al, 2011 and Klug et al 2013). It is unclear whether Ubc9-K157 was deleted to prevent E2 sumoylation or is there any other reason for using K153R? This should be clearly stated. If the E2 still can be modified in the reactions, reducing and non reducing conditions need to be shown in single turnover assays.
5. The substrate coil in Figure 2 represents Smc5-Nse2 automodification. Is there any evidence that only Smc5 but not Nse2 is modified?
6. Multiturnover assay with human E1, E2 enzymes and SUMO1: Smc5-Nse2 efficiently assembles SUMO1 chains. Just out of curiosity, have the authors tested also SUMO2/3?
7. By showing the RAW figures in Suppl Figures it would be helpful to refer to the main Figure and to use the same labelling.

Reviewer #2 (Remarks to the Author):

In this manuscript, Varejão et al. reported the crystal structure of a yeast SUMO E3 ligase, Nse2/Smc5, bound to E2-SUMO thioester mimetic and a non-covalent SUMO. The structure reveals how SP-RING domain and two SIM-like motifs in Nse2 bind E2-SUMO and non-covalent SUMO. The interactions observed in the structure were validated with mutagenesis analysis, kinetic analysis and phenotypic assays in yeast. The data demonstrated that SP-RING domain and SIM motif 1 bind E2 and SUMO portion of E2-SUMO, respectively, and arrange E2-SUMO in a closed conformation to promote Nse2-mediated SUMO conjugation. The C-terminal region of Nse2 harbors a SIM motif 2, which interacts with the non-covalent SUMO bound at the backside of E2. This SIM2-SUMO interaction contributes to the

catalytic efficiency of Nse2/Smc5 SUMO conjugation activity. Overall the data are solid and the findings advance the understanding of how different SUMO ligases organize E2-SUMO and a non-covalent SUMO to enhance the SUMO ligase activity.

I only have few comments.

1. Figure 1B, the short Nse2/Smc5 was slightly less active than the WT, can the authors speculate how the missing regions might affect the activity?
2. In line 313-320, the authors stated that ssDNA increases substrate affinity. Does C-terminal p53 domain used in the assay bind to Nse2/Smc5 or does it contain a SUMO consensus motif and is recognized by Ubc9 directly? If latter, can the authors speculate how ssDNA increases substrate affinity?
3. Line 324-325, this reviewer is unclear on this statement on how gel filtration chromatography (Figure 5B) could show increased stability with the fused Nse2-SUMOB compared to WT Nse2. Figure 5B showed that E2-Smt3 eluted earlier. This needs to be explained.
4. The authors pointed out that SIM2 is only present in yeast Nse2, but not in human. Can the authors comment on how this difference could influence human Nse2? Could this SIM motif present in other component within the Smc5/6 complex?

Minor comments

1. Please define SUMOD and SUMOB at first instance.
2. "E2" was used frequently in the results section rather than Ubc9. It would be useful to state that Ubc9 is the only SUMO E2 and hereafter Ubc9 will be referred to as E2.
3. Would be useful to state Nse4 was used as the substrate in line 147-148 when referring to Figure 1B.
4. Figure 6 panel C is labelled incorrectly.
5. Line 413-414, phosphorylated tyrosine is before the RING domain.

Reviewer #3 (Remarks to the Author):

The SUMO/Smt3 conjugation pathway consists of very few enzymes, compared to the ubiquitin conjugation pathway, including a small number of SUMO/Smt3 E3 ligases. Here the authors report the structure of the Nse2 Smt3 E3 ligase complex, described at atomic resolution. Two Smt3-Interaction Motifs (SIMs) in Nse2 are identified that undergo restructuring during Smt3-binding, contribute to E3 ligase activity and are critical for responses to DNA damaging agents when combined with other mutants, correlating structure and function of these SIMs in an excellent manner. These SIMs in Nse2 were missed in competing papers (Yu et al 2021 PNAS and Taschner et al. 2021 EMBO J), therefore the current paper has sufficient novelty compared to these competing papers. Two issues need to be addressed as detailed below.

1. The SIM mutants are constructed in a rather unusual manner. To demonstrate the true nature of these SIMs, simultaneously mutating their large hydrophobic [VIL] residues to alanine to prevent interaction with Smt3 would be important.

2. The authors hypothesize in the discussion (lines 495-501) that ATR-mediated phosphorylation of Ser260 and Ser261, located next to SIM2 probably enhances Nse2 ligase activity. This hypothesis is worthwhile testing to strengthen the signaling aspect of the paper.

REVIEWER COMMENTS

Reviewer #1 (Remarks to the Author):

To catalyze the SUMO discharge from the E2 to the substrate, bona fide SUMO E3 ligases orient the donor SUMO via a SUMO interaction motif (SIM) in a closed conformation. A second SIM that binds a scaffold SUMO in the backside of Ubc9 stabilizes the E2-E3 complex. This scaffold SUMO is essential for the ZNF451 family and improves Siz/Pias (SP) ligase activity. However, the SIMs mapped for the Siz1 are not conserved in Nse2/Mms21, raising the question how these enzymes execute E3 ligase function. The study proposed by Nathalia Varejao and colleagues addressed this important topic by solving the crystal structure of Nse2-Smc5 in complex with a SUMO charged E2 mimetic and one additional SUMO. Structural, biochemical and cellular analysis disclosed that also in this case two SIMs have key functions for Nse2 E3 ligase activity. Analog to the other SUMO E3 ligases, one SIM positions the donor SUMO, while the second SIM stabilizes the E2/E3 complex by sandwiching a scaffold SUMO on the backside of the E2. Furthermore, the role of ssDNA that was earlier shown to improve E3 ligase function was further analyzed. Provided data indicate that ssDNA rather increases the K_m than the K_{cat} . Overall, the proposed work provides important insights and should be published in a journal like Nature Communications, but needs revision for publication.

Major comments

1. Figure 1B. The authors compare the activities of the Smc5/Nse2_{short} with Smc5/Nse2 WT and describe that their activities are comparable and substantially not affected. Although, Figure 1B shows that Smc5/Nse2_{short} is active, it is significantly reduced compared to WT. This needs rephrasing.

R. We have modified in the main text the sentence regarding this point. The global SUMOylation activity has decreased, in particular regarding high molecular species compared to the single cNse4-SUMO substrate. This could be a consequence of missing lysine acceptors in the shorter version of the Smc5/Nse2.

In any case, we have rephrased in the main text (line 128) to: *“The E3 ligase activity of the Smc5/Nse2_{short} is comparable to the WT Smc5/Nse2, at least for the single SUMO conjugation to cNse4 (cNse4-SUMO), only high molecular conjugates are substantially diminished”*

2. Figure 2 E-G. To draw clear conclusions about enzyme functions it is important to discriminate between substrate modification, Smt3-chains and modification of the enzymes in the in vitro reactions. All SUMO enzymes are efficiently modified in in vitro assays, especially E3 ligases are highly autosumoylated and assemble free or attached SUMO chains.

R. As the reviewer points out, we are aware that all the enzymes used in this assay can be themselves targets for SUMOylation. In face of the clear difficulty into discriminate between different conjugated substrates bands, we focused on measure the impact of Nse2-SIM by quantifying only the first SUMOylated band of those referred model substrates (cp53-SUMO, cNse4-SUMO, coil-SUMO). So, we have modified the labels of

the y-axes of the error bar graphs in Figure 2, Figure 6, Supp. Fig. 2, and *new* Supp. Fig.7 for clarity.

This reviewer is aware that enzymes and substrates in this study share the same tag and likely for the proteins domains used as model substrates no specific antibodies are available. Preferable, one substrate specific analysis should be included. If this is not feasible, at least minus substrate controls have to be included.

R. Indeed, Figure 2G corresponds to *minus substrate* (since no external substrate such as cp53 and cNse4 was added). We don't have specific Ab for each protein, but we have added MW markers to better identify the different SUMOylated species.

Additionally, *just for the reviewer*, we have also run the reactions side by side between the three substrates to discriminate between the different SUMOylated species. At least at low molecular weight (between 15 – 70 KDa), comparison between the three substrates facilitate the identification of the SUMOylates species (as labelled in Figure 2).

3. All substrate sumoylation assays presented show that substrates are modified with multiple Smt3s. Is it understood, whether these model substrates are modified at multiple sites, with Smt3-chains or a combination of both?

R. We understand that substrates can be modified by a combination of multiple sites, and, also by Smt3-chains. For example, our MS data suggest cNse4 can be sumoylated at different lysines, but the most abundant are Lys266 and Lys271.

To these reviewer's understanding, cp53 has only one SUMO site but forms tetramers.

R. Certainly, the cP53 tetramerization domain can only be SUMOylated on Lys386, but it can form SUMO chains under certain conditions.

As an example, on the gel depicted here, SUMO chains on the P53 substrate can be observed (*just for the reviewer's assessment*):

The E2~Smt3 thioester is indicated in Figure 2E, suggesting a multiturnover reaction under non-reducing conditions was performed. This raises the question whether all bands represent conjugates or also protein-complexes are detected?

R. Since only Smt3 is labelled with Alexa-fluorophore, only SUMO conjugated species can be detected. However, we cannot discard in non-reducing gels the presence of a disulphide-crosslinked proteins in the high molecular weight region, which could represent the presence of protein complexes. However, under SDS-denaturing conditions complexes are disassemble and the presence of crosslinked protein complexes must be minimal.

In case this assumption is correct, samples separated upon reducing conditions would clarify this issue.

R. In the case of cp53 substrate (Fig 2e), we have selected non-reducing gels just to show the presence of E2-SUMO thioester, but we can clearly distinguish the first and second SUMO attached conjugates to cp53, which are our bands of interest used in the quantification of the plots.

Here in this gel, *just for the reviewer's assessment*, the multiple turnover reaction under denaturing condition displays the SUMO-conjugated bands generated for the cp53 substrate, which are similar to the bands displayed in Figure 2e.

4. It is noticeable that the work of the Pichler lab is explicitly not cited. Is there a reason for this? The Pichler lab has identified and characterized the two SIM dependent mechanism for the SUMO E3 ligase ZNF451 (Eisenhardt et al, 2015, Koidl et al 2016) and proposed that a similar mechanism likely applies for SP-RING E3 ligases (Pichler et al, 2017). This idea was even adapted in a recent review from the Reverter lab (Varejao, 2018), seemingly they are aware of these studies. The Pichler lab has further shown that the backside of Ubc9 contributes to SUMO chain assembly involving yeast Ubc9 sumoylation or a scaffold SUMO bound by ZNF451 (Klug et al, 2013, Eisenhardt et al, 2015, Pichler et al, 2017).

R. Sorry for this unintended mistake. We deeply apologize for this misunderstanding, it has been unintentional on our side, since we thought that *Capadoccia et al., 2015* included all those points, and that was our misinterpretation. We have added all missing references in the text (in the introduction, results and discussion), for both the role of the two SIMs in the ZNF451 E3 ligase and the role of ZNF451 in the SUMO chain assembly. Finally, we greatly acknowledge the contribution of the Pichler lab in those discoveries on the E3 ligase mechanisms.

As Figure 2 demonstrates that poly-Smt3 conjugates drastically disappear in various Nse2 mutations targeting SIM2 but less severely by Nse2 mutations targeting SIM1 or the Nse-E2 binding interface (Figure 6) raises the intriguing question whether SP-RING E3 ligases also assemble SUMO chains via the E2 backside.

R. In Figure 2f and in Figure 6d, similar multiple turnover experiments were conducted with the cNse4 substrate. In both instances a similar decrease of PolySUMO is observed, as depicted in the following gels from figs 2 and 6 (shown below).

Bands marked in Figure 2 as Poly-Smt3 can, as pointed out above, present substrate and enzyme modifications, attached and free SUMO chains.

R. This is correct. We would like to clarify that we used “PolySUMO” to refer (in a general manner) all those possibilities cited by the reviewer.

Many of these bands disappear even after 5 minutes reaction time in the presence of Nse2delta4 and a substrate (cNse4 and cp53) but to a lesser extent when automodification/coil is tested.

R. From our understanding of the results, when using the automodification/coil substrate, we also observed that polySUMO disappears with the SIM2 mutants, similarly to cNse4 and cp53. Please see the gel from Figure 2g (shown here).

These data suggest that some fraction of the high molecular smear represents E3 automodification and the bands that remain unchanged, likely present E1 (Aos1 and Uba2) and E2 modifications. This is correct. We just chose not to discriminate between them. E3 automodification is a cis-reaction and appears to depend less on SIM2, compared to trans-reactions required to modify the substrate.

R. This is a good point that we don't really know for sure. However, in our in vitro reactions containing a high amount of E3 (for example 400nM for multiple turnover with the coil substrate, figure 2g), a probable increase of SUMOylation in trans would be plausible (between different molecules in the reaction mixture), in which the coil-Nse2/Smc5 could also act as a substrate.

Further support for a function of the E2-backside SUMO-interaction in chain formation is indicated in Figure 4 D demonstrating that c-Nse4-2Smt3 and cp53-2Smt3 conjugates disappear in the presence of 1.5 x non-conjugateable Smt3 D68R, while at longer time points even in the reactions lacking the E3 (later time points) cp53-2Smt3 was detected.

R. We agree with the reviewer point of view, and we have rephrased a sentence in the main text to point out the importance of back-side on polychains formation.

Sentence (line 248-250): *"In vitro conjugation reactions with Smt3 D68R shows a strong decrease in the conjugation rates, particularly for polySUMO chains, either in the presence or absence of the E3 ligase (Fig. 4c)".*

5. E2 discharge reactions in Figure 4: In the Figure legend it is stated "D68RSUMO is activated by E1 and transferred to the E2 in similar rate as the wild type (left)." The figure shown presents a significant reduced charging of the E2 with SUMO D68R! This needs rephrasing.

R. Sorry for that. We have re-written this sentence (line 821-822): *"^{D68R}SUMO is activated by E1 and transferred to the E2 in similar at lower rates as the wild type (left)".*

However, in the discharge assays it appears that similar amounts of charged E2s were used. The discharge assays to cp53 but not to cNse4 detects the charged E2 enzyme to follow its discharge. Is the cNse4 gel separated under reducing conditions or why is the charged E2 not detected in this case?

R. Sorry for the misunderstanding. In the case of cNse4 the gel was run under reducing conditions, because the size of E2~Smt3 thioester is very similar to the cNse4-Smt3 and the E2-Smt3 thioester cannot be observed. We have re-labelled below the gels in Fig 4d the oxidative condition of the samples in the SDS gel.

6. E2 discharge reactions in Figure 5 display very weak discharge compared to Figure 4. What is the reason for that? According to the understanding of this reviewer the main difference in the reactions is the use of Smt3-E3 fusions in Figure 5 compared to addition of free Smt3 in Figure 4.

R. The reviewer's observation is correct. We also noticed that. We attribute this difference to the presence of Sodium citrate in the reaction present in Figure 5 (affecting both, non-fusion and fusion E3, Fig.5a,e). This molecule was used as a buffer in the purification of E2~Smt3 thioester, as previously used elsewhere (Cappadocia, 2015; Streich and Lima, 2016). Actually, we ended up taking advantage of the reaction slowdown making it possible to calculate the slope of the reactions (V_0) and the differences among the different constructs analyzed. We are currently investigating the effect of citrate in the discharge reaction catalyzed by Nse2/Smc5 E3-ligase.

Considering the corresponding raw data shown in Figures S4 and S5, indicate that reactions containing free Smt3 allow poly-SMT3 conjugates that are not detected in any reaction using the Smt3-E3 fusion.

This raises the question, whether the nature of these poly-Smt3 conjugates is E3 automodification that is blocked upon Smt3-E3 fusion (see also below)? Polysumoylation of cp53 is unlikely as it has only one sumoylation site and chain formation should be largely impaired by using Smt3D68R.

R. This is a good point that should be investigated in the future. We do believe that poly-Smt3 in these gels is a result of Smc5/Nse2 automodification, which also happens when

we use Smt3-E3 fusion at the same concentration used for non-fusion E3 in Figure 4 (400 nM).

Just as an example, in this gel (*only for the reviewer's assessment*), SUMO chains can also be formed efficiently and fast with the E3-SUMO fusion, but only when the buffer condition is optimal and the concentration of the E3 is high.

7. Substrate E3 interaction. To this reviewers understanding, cp53 is a model substrate that is unlikely to directly interact with Smc5/Nse2short. The cp53 fragment displays the tetramerization domain and one sumoylation site but lacks the DNA binding domain. As the presented data indicate that ssDNA rather induces the K_M than the K_{cat} , this raises the question how cp53 interacts with the E2-E3 ligase complex. A low affinity interaction between cp53's SUMO consensus site and the catalytic cleft of the E2 is likely (Bernier-Villamor V et al, 2002) and sufficient for modification by the E2-E3 complex (Figure 2 E, left panel), especially at the high substrate concentrations used. As cp53 lacks its established DNA binding domain the question that arises is whether cp53 is able to interact with DNA.

R. We agree with the that the cp53 C-terminal tetramerization domain is unlikely to specifically interact with DNA in the absence of the DNA-binding domain. However, this raises the question about the effect on the K_M of the reaction. We also agree that the interaction between cp53's SUMO consensus site and the catalytic cleft of the E2 would be sufficient for SUMO conjugation. We believe that DNA binds the Smc5 coil region, that is connected to the Nse2 E3 ligase and in turn might improve its binding to the E2-SUMO thioester (a second half of the enzymatic substrate) (Varejao et al., 2018). But it is unlikely that DNA directly binds the cp53 tetramerization domain alone.

Alternatively, Figure S4 suggests that ssDNA reduces E3 automodification to the benefit of substrate modification? In the absence of ssDNA, poly-SMT3 appears to increase compared to the ssDNA containing reactions. Is it possible, that E3 automodification competes with substrate modification and the E3 bound to ssDNA reduces automodification to the benefit of substrate modification? Titration of ssDNA to the reactions and discrimination between substrate and E3 sumoylation would clarify this issue.

R. We agree that there could be a competition between E3 automodification and substrate modification, and the ssDNA might be involved. However, this is difficult to check with the gels that we have run, and this was not a major aim of our work. We were basically interested in the role of SIM2 in SUMOylation, and we have mainly checked the single and double bands of the cp53 and cNse4 SUMO conjugates.

In any case, in a previous work (Varejao *et al.*, 2018), we already run a titration of Smc5/Nse2 with different concentrations of ssDNA. In any case we can observe that binding of ssDNA to Smc5 does not impede the formation of polySUMO chains. Here we are displaying figure 1c from that work [10.15252/embj.201798306](https://doi.org/10.15252/embj.201798306).

[redacted]

Minor comments:

1. Smt3 and SUMO are used to describe the yeast SUMO, Smt3. For clarification, a uniform labeling would avoid confusion: one could either use Smt3 throughout the manuscript or probably ySUMO or S.c.SUMO.

R. We have added in the main text (line 115-117) the sentence: *“To facilitate comprehension, we will use SUMO to refer to yeast Smt3 and E2 to yeast Ubc9 throughout the text.”*

2. All gel presentations require size markers.

R. Sorry for that. The size markers do not appear because the gels are photographed by fluorescence. To satisfy the reviewer request, we are labeling the bands with their reasonable molecular weight.

3. For clarification, reducing and non-reducing gels need to be indicated throughout the manuscript. It would also be helpful for the reader to state all enzyme concentrations in each Figure.

R. We have added below all SDS-gels the different type of buffer oxidative condition for clarification.

4. Ubc9-K153R is used for structural analysis and E2 discharge experiments, likely to prevent E2 sumoylation. However, the yeast Ubc9 is modified at K153 and K157 (Ho *et al.*, 2011 and Klug *et al.* 2013). It is unclear whether Ubc9-K157 was deleted to prevent E2 sumoylation or is there any other reason for using K153R? This should be clearly stated.

R. The point of the reviewer is correct. We followed the same protocol described by Streich and Lima (2016,2018), in which only K153R was mutated. In fact, during the purification of E2-SUMO mimetic for complex crystallization, we set up a method to separate single from multiple SUMO-E2 modifications by two steps of gel filtration and ion exchange chromatography.

We have now clearly stated this point in the main text (line 117-119): “Such E2-SUMO thioester mimetic has been engineered based on previous work (Streich and Lima, 2016, 2018), by the substitution of Ala129 to lysine in a location next to the active site Cys93 in Ubc9, and Lys153 for arginine to prevent unwanted E2 SUMOylation (Ho et al, 2011 and Klug et al 2013).”

If the E2 still can be modified in the reactions, reducing and non-reducing conditions need to be shown in single turnover assays.

In our in vitro conjugation reactions we have used Ubc9 K153R to prevent unwanted E2 SUMOylation (as mentioned in the materials and methods section), but we cannot exclude the possibility of SUMO conjugation in K157.

As observed from this figure (for reviewer's assessment) E2-SUMO thioester reaction also displays additional bands in the non-reducing gel (time 0). Some of these bands are still present under reducing conditions, indicating that they could be SUMOylated forms of the E2, but they are very residual.

5. The substrate coil in Figure 2 represents Smc5-Nse2 automodification. Is there any evidence that only Smc5 but not Nse2 is modified?

R. We agree with the reviewer point and we cannot discard Nse2 SUMOylation, as shown in the this report (<https://www.ncbi.nlm.nih.gov/pmc/articles/PMC538766/>)

6. Multiturnover assay with human E1, E2 enzymes and SUMO1: Smc5-Nse2 efficiently assembles SUMO1 chains. Just out of curiosity, have the authors tested also SUMO2/3?

R. Yes. We have tested the reaction with SUMO2/3 and it can also be SUMOylated.

7. By showing the RAW figures in Suppl Figures it would be helpful to refer to the main Figure and to use the same labelling.

R. As suggested by the reviewer we have added the main text reference in the Suppl. Figures to better facilitate the comprehension of the reader.

Reviewer #2 (Remarks to the Author):

In this manuscript, Varejão et al. reported the crystal structure of a yeast SUMO E3 ligase, Nse2/Smc5, bound to E2-SUMO thioester mimetic and a non-covalent SUMO. The structure reveals how SP-RING domain and two SIM-like motifs in Nse2 bind E2-SUMO and non-covalent SUMO. The interactions observed in the structure were validated with mutagenesis analysis, kinetic analysis and phenotypic assays in yeast. The data demonstrated that SP-RING domain and SIM motif 1 bind E2 and SUMO portion of E2-SUMO, respectively, and arrange E2-SUMO in a closed conformation to promote Nse2-mediated SUMO conjugation. The C-terminal region of Nse2 harbors a SIM motif 2, which interacts with the non-covalent SUMO bound at the backside of E2. This SIM2-SUMO interaction contributes to the catalytic efficiency of Nse2/Smc5 SUMO

conjugation activity. Overall the data are solid and the findings advance the understanding of how different SUMO ligases organize E2-SUMO and a non-covalent SUMO to enhance the SUMO ligase activity.

I only have few comments.

1. Figure 1B, the short Nse2/Smc5 was slightly less active than the WT, can the authors speculate how the missing regions might affect the activity?

R. We agree with the reviewer. Nine lysine residues are deleted in Nse2 and seven in Smc5 in the shorter versions, which could be additional sites for poly-SUMOylation by automodification of the E3 ligase.

We have modified in the main text the sentence regarding this point. The global SUMOylation activity has decreased, in particular regarding to high molecular species compared to the single cNse4-SUMO substrate.

In any case, we have rephrased in the main text (line 128-130) to: *“The E3 ligase activity of the Smc5/Nse2_{short} is comparable to the WT Smc5/Nse2, at least for the single SUMO conjugation to cNse4, only high molecular conjugates are substantially diminished”*

2. In line 313-320, the authors stated that ssDNA increases substrate affinity. Does C-terminal p53 domain used in the assay bind to Nse2/Smc5 or does it contain a SUMO consensus motif and is recognized by Ubc9 directly? If latter, can the authors speculate how ssDNA increases substrate affinity?

R. Regarding the point of DNA and the substrate affinity. The interaction between cP53's SUMO consensus site (*Bernier-Villamor et al, 2002*) and the catalytic cleft of the E2 is sufficient for the SUMO modification by the E2-E3 catalytic module. We previously showed that DNA binds the Smc5 coil region reducing the K_M of the reaction (*Varejao et al., 2018*). Since the Smc5 coil is connected to the Nse2 E3 ligase forming a complex, DNA binding to Smc5 might in turn improve binding of the E2-SUMO thioester (the second half of the substrate of this enzymatic reaction) by a structural rearrangement of the Smc5/Nse2 complex (*Varejao et al., 2018*). But it is unlikely that DNA directly binds the C-terminal tetramerization domain of P53.

3. Line 324-325, this reviewer is unclear on this statement on how gel filtration chromatography (Figure 5B) could show increased stability with the fused Nse2-SUMOB compared to WT Nse2. Figure 5B showed that E2-Smt3 eluted earlier. This needs to be explained.

R. Sorry for that misunderstanding, this is a point that needs a better explanation. In the complex between the E2-thioester mimic and the E3-SUMO fusion complex, the complex formation can be experimentally observed in the gel filtration by a single peak including both components. However in the gel filtration with the non-fused form of the E3, such a peak containing the complex is not observed, probably indicating a weaker binding between the E3 and the E2-SUMO thioester compared to the complex with the E3-SUMO fusion.

Also, the elution profile of the E2-SUMO thioester mimetic in the gel filtration is very unusual, and comes off the column as a high molecular mass complex, far from its theoretical elution volume.

We have added a sentence in the main text (line 290): *“Gel filtration chromatography showed an increased stability of the complex formed between the E2-SUMO mimetic and the fused Nse2-SUMO_B, compared to WT-Nse2, which did not elute as a single peak (Fig. 5d).”*

4. The authors pointed out that SIM2 is only present in yeast Nse2, but not in human. Can the authors comment on how this difference could influence human Nse2? Could this SIM motif present in other component within the Smc5/6 complex?

R. As pointed out by the reviewer, it is true that SIM2 is only present in the yeast system. In humans we could speculate about an alternative mechanism to fix SUMO at the E2 backside. For example, SUMOylation of other components of the Smc5/Smc6 complex, such as the long coiled-coil region of Smc5 that has been recently shown to be SUMOylated (<https://doi.org/10.1016/j.celrep.2019.10.123>), could fix SUMO to the backside of E2 and enhance the E3 activity by a similar mechanism, but not dependable on the C-terminal tail of Nse2.

At the end of the discussion (last paragraph, line 456), we hypothesize on that issue.

Minor comments

1. Please define SUMOD and SUMOB at first instance.

R. SUMOD is donor SUMO and SUMOB is the SUMO binding the E2 backside. We have now defined these terms in the introduction section (lines 101 and 104).

2. “E2” was used frequently in the results section rather than Ubc9. It would be useful to state that Ubc9 is the only SUMO E2 and hereafter Ubc9 will be referred to as E2.

R. We have added this sentence in the first paragraph of the results section (line 115-117): *“To facilitate comprehension, we will use SUMO to refer to yeast Smt3 and E2 to yeast Ubc9 throughout the text.”*

3. Would be useful to state Nse4 was used as the substrate in line 147-148 when referring to Figure 1B.

R. Thanks for this observation. We have change this sentence for (line 128-130): *“The E3 ligase activity of the Smc5/Nse2_{short} is comparable to the WT Smc5/Nse2, at least for the single SUMO conjugation to cNse4 (cNSE4-SUMO), only high molecular conjugates are substantially diminished”*

4. Figure 6 panel C is labelled incorrectly.

R. Sorry for that. We have corrected this mistake.

5. Line 413-414, phosphorylated tyrosine is before the RING domain.

R. Sorry for this mistake. We have modified the sentence, and replaced *after* for *before* (line 378).

Reviewer #3 (Remarks to the Author):

The SUMO/Smt3 conjugation pathway consists of very few enzymes, compared to the ubiquitin conjugation pathway, including a small number of SUMO/Smt3 E3 ligases. Here the authors report the structure of the Nse2 Smt3 E3 ligase complex, described at atomic resolution. Two Smt3-Interaction Motifs (SIMs) in Nse2 are identified that undergo restructuring during Smt3-binding, contribute to E3 ligase activity and are critical for responses to DNA damaging agents when combined with other mutants, correlating structure and function of these SIMs in an excellent manner. These SIMs in Nse2 were missed in competing papers (Yu et al 2021 PNAS and Taschner et al. 2021 EMBO J), therefore the current paper has sufficient novelty compared to these competing papers. Two issues need to be addressed as detailed below.

1. The SIM mutants are constructed in a rather unusual manner. To demonstrate the true nature of these SIMs, simultaneously mutating their large hydrophobic [VIL] residues to alanine to prevent interaction with Smt3 would be important.

R. This is a good observation. To mutate the Nse2 SIM2, we have considered and prepared different combinations (see Supplementary Fig. 2). We designed some of the SIM2 mutants before knowing the complex structure, in which Ile264 and Val266 are inserted in the hydrophobic SUMO surface cavity. Mutation of Ile264 for Pro would disrupt the extended beta conformation of SIM, and mutation of Val266 for Arg would induce important steric clashes. We also prepared a double mutant Asp265Ala/Leu267Ala, which is more canonical, and the SIM deletion mutant (Delta4Nse2). In all instances the reduction of the SUMO conjugation activity is quite similar, indicating that all of them perturb the interface (see Suppl. Fig. 2).

2. The authors hypothesize in the discussion (lines 495-501) that ATR-mediated phosphorylation of Ser260 and Ser261, located next to SIM2 probably enhances Nse2 ligase activity. This hypothesis is worthwhile testing to strengthen the signaling aspect of the paper.

R. This is a good point raised by the reviewer. To check this hypothesis, we have prepared the following Nse2 mutants: S260E, S261E and S260E/S261E double mutant; and run in vitro SUMO conjugation reactions. The results are displayed in a new Supplemental Figure (Supplementary Fig. 7) and at least, in our in vitro conditions, the differences with the wild-type are not significant, but we cannot discard the presence of a PhosphoSIM in vivo.

We have included a sentence at the end of the discussion (line 447): *“Despite our in vitro SUMO reactions with phosphomimetic S260E, S261E and S260E/S261E Nse2 constructs do not display a substantial SUMO conjugation increase compared to WT (Supplementary Fig. 7), we cannot discard that SIM2 may be a Phospho-SIM in vivo, further enhancing SUMO ligase activity under conditions that activate the DNA damage checkpoint.”*

REVIEWERS' COMMENTS

Reviewer #1 (Remarks to the Author):

I have no further concerns and support publication.

Reviewer #2 (Remarks to the Author):

The authors have addressed all my concerns. The manuscript is suitable for publication.

Reviewer #3 (Remarks to the Author):

The authors have properly addressed my concerns.